# A parallel implementation of the confined-unconfined aquifer system model for subglacial hydrology: design, verification, and performance analysis (CUAS-MPI v0.1.0)

Yannic Fischler[1], Thomas Kleiner[2], Christian Bischof[1], Jeremie Schmiedel[4], Roiy Sayag[4], Raban Emunds[1,2], Lennart Frederik Oestreich[1,2], and Angelika Humbert[2,3]

[1]Department of Computer Science, Technical University Darmstadt, Darmstadt, Hesse, Germany
[2]Alfred-Wegener-Institut, Helmholtz-Zentrum für Polar- und Meeresforschung, Bremerhaven, Bremen, Germany
[3]Faculty of Geosciences, University of Bremen, Bremen, Germany
[4]Department of Environmental Physics, BIDR, Ben-Gurion University of the Negev, Sde Boker, Israel

**Correspondence:** Yannic Fischler (yannic.fischler@tu-darmstadt.de)

**Abstract.** The subglacial hydrological system affects (i) the motion of ice sheets through to sliding, (ii) the location of lakes at the ice margin, as well as (iii) the ocean circulation by freshwater discharge directly at the grounding line or (iv) via rivers flowing over land. For modelling this hydrology system, a previously developed porous medium concept called Confined-Unconfined Aquifer System (CUAS) is used. To allow for realistic simulations at the ice sheet scale, we developed CUAS-MPI, an MPI-parallel C/C++ implementation of CUAS, which employs the PETSc infrastructure for handling grids and equation systems. We validate the accuracy of the numerical results by comparing them with a set of analytical solutions to the model equations, which involve two types of boundary conditions. We then investigate the scaling behavior of CUAS-MPI and show that CUAS-MPI scales up to 3840 MPI processes running a realistic Greenland setup on the Lichtenberg HPC system. Our measurements also show that CUAS-MPI reaches a throughput comparable to that of ice sheet simulations, e.g. the Ice-sheet and Sea-level System Model (ISSM). Lastly, we discuss opportunities for ice-sheet modelling, future coupling possibilities of CUAS-MPI with other simulations, and consider throughput bottlenecks and limits of further scaling.

## 1 Introduction

The dynamics of ice sheets in Greenland and Antarctica is highly related to the conditions at the ice base (e.g. Engelhardt and Kamb, 1997; Bindschadler et al., 2003; Hoffman et al., 2011; Siegert, 2000). At the base of Greenland, the ice is over at least 33% thawed (MacGregor et al., 2022) at the pressure melting point, with melt rates reaching as much as $0.19\,\mathrm{m\,a^{-1}}$ in the inland (Zeising and Humbert, 2021) and up to $15\,\mathrm{m\,a^{-1}}$ in outlet glaciers (Young et al., 2022). Hence, an extensive subglacial hydrological system is expected to exist. In addition, the margins of the Greenland Ice Sheet are experiencing massive surface melt in summer (Colosio et al., 2021), which is partially stored in supraglacial lakes (Schröder et al., 2020). Most of them drain

eventually and deliver the water to the subglacial hydrological system rapidly (Neckel et al., 2020). As the subglacial system is hidden beneath ice hundreds to thousands meters thick, observations are extremely sparse and establishing a representative mathematical model is challenging.

Understanding of glacier hydrology has been developed in the past century mainly by investigating mountain glaciers. Comprehensive overviews are given by (Fountain and Walder, 1998) and Flowers (2015) including the involved processes,

observational evidence and numerical modelling. The need to incorporate the effect of subglacial hydrology on the lubrication of ice masses inspired so-called flux-routing (or balance-flux) schemes, which model the hydraulic potential assuming time-invariant steady-state water pressures derived from the ice-overburden pressures (e.g. Budd and Jenssen, 1987; Le Brocq et al., 2009). Due to their simplicity, such thin water sheet models are still in use (e.g. Franke et al., 2021), despite their limitations in representing the system, most notably their inability of switching between distributed (inefficient) and channelised (efficient)

water transport (Figure 1). As they also do not include the water pressure, workarounds are needed to use them in sliding laws of ice sheet models (Kleiner and Humbert, 2014). A more advanced approach to simulate the inefficient system was made by (Bueler and van Pelt, 2015), which simulate the subglacial hydrology based on Darcy-type flow reformulated into an advection–diffusion-production equation for the evolution of the conserved water amount. An early attempt to include the effective pressure from both types, channels and cavities, into large-scale ice sheet modelling was presented by Arnold and

Sharp (2002) based on the work of (Fowler, 1987) and (Lliboutry, 1978).

A major step in the formulation of subglacial hydrology models was the generalisation of the system into a porous medium sheet (e.g. Flowers and Clarke, 2002; Hewitt, 2011) resembling a multi-component hydrological system. This development resulted in more advanced mathematical models, including partial differential equations, with new numerical and computational demands. The models have in common that they solve a diffusion equation (either for the head or water layer thickness)

and that they represent the components of the hydraulic system by evolving pore space, thus the geometry of the drainage space (typically opening by melt and/or ice flow, and closure by ice creep). Werder et al. (2013) developed a model that uses unstructured meshes with channels at element edges and distributed flow within the element with water exchange between the two components. De Fleurian et al. (2014) use a double continuum approach with two different porous layers, one for the distributed system and one for the efficient system, with the second being important for the seasonal evolution of the

hydrological system (de Fleurian et al., 2016). While in these approaches different sets of governing equations are given for the different systems, other approaches rely on one set of governing equations for both systems. Sommers et al. (2018) evolves the water layer thickness and allows for a transition between laminar and turbulent flow by evaluating the local Reynolds number. Beyer et al. (2018) is based on Darcy flow, evolves the transmissivity and introduces a confined-unconfined aquifer scheme.

Some of these models are incorporated into ice sheet models or coupled to ice sheet models, which led to the first coupled

simulations of ice-sheet-hydrology for individual ice stream/glacier systems with advanced hydrology models, such as in Smith-Johnsen et al. (2020), Cook et al. (2020, 2022) and Ehrenfeucht et al. (2023). However, future projection simulations for Greenland (Goelzer et al., 2020) or Antarctica (Seroussi et al., 2020) do not yet include any coupled ice-sheet-hydrology models.

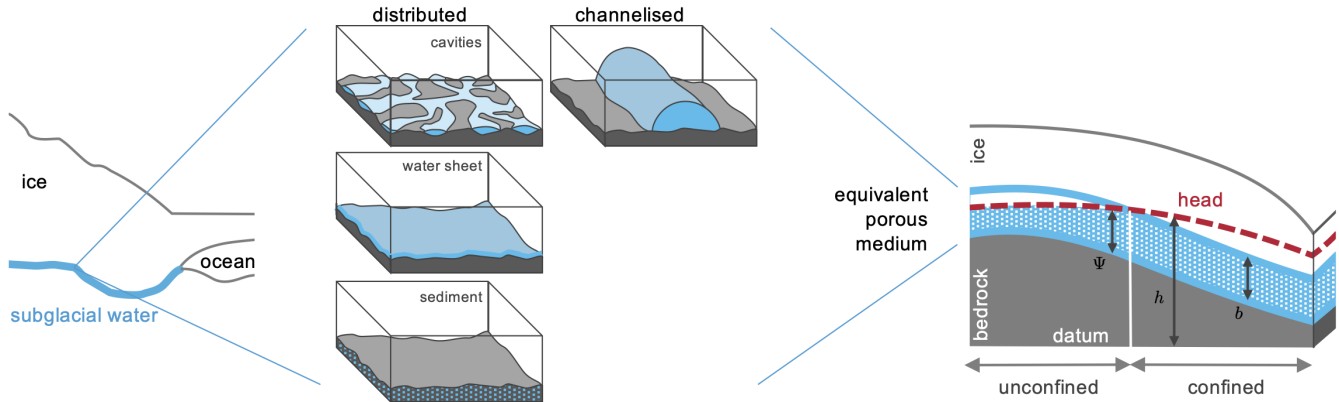

**Figure 1.** Modelling concept of CUAS-MPI. The left side shows the ice sheet and the underlying subglacial system. In the middle the different forms of the subglacial system are shown (after (Benn and Evans, 2010)). The right part represents the equivalent porous medium approach and selected variables of the model (Section 2.1).

Our own work is heavily inspired by the equivalent porous media approach (EPM) for subglacial hydrology published by de Fleurian et al. (2014, 2016) which led to the development of the Confined-Unconfined Aquifer System model (CUAS, Beyer et al., 2018) and its prototype implementation written in the Python language. This EPM approach offers a numerically relatively inexpensive way of incorporating different elements of the hydrological system, such as a thin water sheet, channels, cavities and water transport within the subglacial sediments. The concept is sketched in Figure 1. Those elements are not resolved individually, but are represented as an effective transmissivity of the equivalent porous medium (right side of Figure 1), hence the ability to transport water.

While the main intention for developing a hydrological model was the influence on the dynamics of the ice sheet via sliding, other disciplines benefit from these simulations, too. Oceanographers are dealing with the simulated water flux across the grounding line or glacier terminus as freshwater input into the ocean/fjord system. Hydrologists need freshwater flux to estimate the discharge available for hydropower as in (Ahlstrøm et al., 2009; Braithwaite and Højmark Thomsen, 1989).

Ice sheet dynamics is contributing significantly to sea level rise, and projections show that this is going to increase in the next centuries (Nowicki et al., 2016; Goelzer et al., 2020; Seroussi et al., 2020). These projections are typically until the year 2100 as climate forcing for such simulations is available for this time period. As the subglacial hydrological system is having a big influence on the sliding of ice over the bed, projections of the contribution of ice sheets to sea level change will benefit from simulations of the subglacial system for the same time period. In order to solve problems such as the projection of the change in the hydrological system until 2100, reflecting the effects of alteration in the seasonal meltwater supply to the base, we need to compute many time steps at fine resolution, in particular around the ice sheet margins, where seasonal meltwater from the ice surface reaches the base. To be able to handle such large systems, we need simulation software capable of using parallel computers efficiently for their aggregate memory and computing power.

Therefore, we developed CUAS-MPI a parallel implementation of the confined–unconfined aquifer scheme (CUAS) presented by Beyer et al. (2018). The new implementation is written in C/C++ employing process parallelism using the Message Passing Interface (MPI) to enable the use of high performance computers. CUAS-MPI allows file in- and output in the Network Common Data Format (NetCDF, Rew and Davis (1990)) and uses the Portable, Extensible Toolkit for Scientific Computation (PETSc, Balay et al. (2021) to handle grids and equation systems. We validate the numerical results of CUAS-MPI with known analytical solutions of pumping tests in hydrology. Specifically, we consider solutions to flows in a confined aquifer setup (Theis, 1935; Ferris et al., 1962) and in an unconfined aquifer setup (Jacob, 1963). We show that the numerical results of CUAS-MPI are consistent with the analytical predictions for both the confined and unconfined cases and for different boundary conditions, indicating an accurate and credible implementation of the numerical scheme for those cases. On a realistic setup of Greenland, we then employed CUAS-MPI with up to 3840 processes, gathering runtime data on relevant code building blocks with Score-P (Knüpfer et al., 2012). The performance results of this model setup demonstrate that the CUAS-MPI software is able to harness the power of parallel computing to enable the simulation of relevant and challenging ice models.

The paper is structured as follows: We explain the underlying model and the software design of CUAS-MPI in Section 2. In Section 3 we describe a pumping test model problem where analytical solutions are known and compare them to the results of CUAS-MPI. We then describe a realistic Greenland setup in Section 4.1 and use this setup in an investigation of the performance and scaling behavior of CUAS-MPI in the remainder of Section 4. Finally we discuss the model and its performance and conclude our work.

## 2 Implementation of CUAS-MPI

### 2.1 Description of the physical model

The subglacial hydrological system is simulated by a single EPM layer of thickness $b$ between two confining layers — the ice sheet and the glacier bed (Figure 1). We use the hydraulic head $h$, also recognized as the piezometric head, to express the water pressure $p_w$, where $h = p_w/(\rho_w g) + z_b$, with water density $\rho_w$, the acceleration due to gravity $g$ and the bed topography $z_b$. The void space in the aquifer is considered saturated, and thus, the water transport can be described by Darcy flow. We use $\Psi = h - z_b$ and distinguish between confined ($b \leq \Psi$) and unconfined ($0 \leq \Psi < b$) conditions. Unconfined conditions may happen if the water supply from, e.g. ice sheet basal melt, is not sufficient to keep the water system fully saturated. In this case, $\Psi$ describes the saturated thickness of the aquifer in which Darcy Flow still applies.

The aquifer is described by its transmissivity $T$, a measure of the rate at which the water can spread. Very efficient water transport (high transmissivity) is thought to take place through a channelised system, while a distributed system is known to lead to inefficient transport (low transmissivity). An increase in $T$ can be thought to be caused by an increase in the number of channels or an increase in the cross-sectional area of existing channels, although we do not distinguish between both in our continuum description. The storativity $S$ is another property of the aquifer and is the volume of water released from storage per unit surface area per unit decrease in the hydraulic head. Switching between the confined and unconfined aquifer conditions is facilitated by the effective variables for storativity, $S_e$, and transmissivity $T_e$. In the following, we summarise the

relevant equations based on the arguments discussed in Ehlig and Halepaska (1976) and Beyer et al. (2018). The symbols and parameters used in the model are given in Table 1.

The vertical integrated mass balance for Darcy systems is given by (Ehlig and Halepaska, 1976):

$$S_{\mathrm{e}}(h)\frac{\partial h}{\partial t} = -\nabla \cdot \mathbf{q} + Q = \nabla \cdot (T_{\mathrm{e}}(h)\nabla h) + Q \tag{1}$$

with the water input $Q$ and the depth integrated water flux $\mathbf{q}$. The effective storativity reads as $S_{\mathrm{e}}(h) = S_{\mathrm{s}}b + S'(h)$ with the specific storage $S_{\mathrm{s}}$ and

$$S'(h) = \begin{cases} 0, & b \leq \Psi \\ (S_{\mathrm{y}}/d)(b - \Psi), & b - d \leq \Psi < b \\ S_{\mathrm{y}}, & 0 \leq \Psi < b - d \end{cases} \tag{2}$$

in which $S_{\mathrm{y}}$ is the yield storage, the ratio of water volume per unit volume which gets released once the aquifer drains. To allow a smooth transition between the confined and unconfined system, the parameter $d$, with $0 \leq d \leq \Psi$, is introduced (Ehlig and Halepaska, 1976). For the experiments presented here, we use $d = 0\,\mathrm{m}$ and, thus, do not apply smoothing in the vertical direction.

The effective transmissivity varies according to

$$T_{\mathrm{e}}(h) = \begin{cases} T, & b \leq \Psi \\ \dfrac{T}{b}\Psi, & b > \Psi. \end{cases} \tag{3}$$

As soon as the head sinks below the aquifer height, only the saturated section contributes to the estimation of the transmissivity. The temporal change of transmissivity is computed by

$$\frac{\partial T}{\partial t} = \frac{g\rho_{\mathrm{w}}KT}{\rho_{\mathrm{i}}L}(\nabla h)^2 - 2An^{-n}|N|^{n-1}NT + \beta|\boldsymbol{v}_{\mathrm{b}}|K, \tag{4}$$

where $K$ is the hydraulic conductivity, $\rho_{\mathrm{i}}$ is the ice density and $L$ is the latent heat of fusion. The first term on the right hand side represents melting, the second term creep opening/closure and the last term the formation of cavities. The creep term incorporates the creep rate factor $A$, the creep exponent $n$ and the effective pressure $N$. The effective pressure is related to the ice overburden pressure $p_{\mathrm{i}} = \rho_{\mathrm{i}}gH$ as $N = p_{\mathrm{i}} - p_{\mathrm{w}}$. Cavity opening is related to the basal ice velocity $v_{\mathrm{b}}$ and a parameter $\beta$ that represents the bed undulation.

We have only two state variables in the model ($h$ and $T$) and all other quantities are either parameters or can be derived from the state variables using equations outlined in Beyer et al. (2018). Boundary conditions are either Dirichlet boundaries with a prescribed head or a no-flow boundaries $T_{\mathrm{e}}\nabla h = 0$ (homogeneous Neumann boundary condition). The later are approximated by setting the $T_{\mathrm{e}} = T_{\mathrm{no\ flow}}$ outside the margin and using the harmonic mean for $T_{\mathrm{e}}$ directly on the boundary (Patankar, 1980). For all ocean grid nodes we use a very high transmissivity $T = T_{\mathrm{e}} = T_{\mathrm{max}}$.

The transient flow equation (Eq. 1) is a two-dimensional diffusion equation with non-uniform and time-dependent hydraulic diffusivity $\alpha(x, y, t) = T_{\mathrm{e}}/S_{\mathrm{e}}$ and a time-dependent source term $Q(x, y, t)$. The equation is discretised spatially on a

135 regular square grid using a second-order central difference scheme (e.g., Ferziger and Perić, 2002). All quantities are co-located on the grid. The implementation allows for first-order approximation (fully-implicit or fully-explicit) and second-order (Crank–Nicolson) approximation in time for the hydraulic head. Nevertheless, only the fully-implicit time stepping is used in this study to make the solution less dependent on the initial conditions for the hydraulic head. The transmissivity is updated using an explicit Euler step. The discretisation of Eq. 1 leads to a system of linear equations. In each time step this system of

140 linear equations is solved using one of the PETSc solvers configured by the user. If an iterative solver is used, convergence is decided by the decrease of the residual norm relative to the norm of the right hand side and the initial residuum (PETSc option `-ksp_converged_use_min_initial_residual_norm`). If the linear system is written as $\boldsymbol{Ax} = \boldsymbol{b}$, where $\boldsymbol{A}$ denotes the matrix representation of a linear operator, $\boldsymbol{b}$ is the right-hand-side vector, and $\boldsymbol{x}$ is the solution vector and $\boldsymbol{r}_k = \boldsymbol{b} - \boldsymbol{Ax}_k$ is the residuum vector of for the $k$-th iteration, then convergence is detected if $||\boldsymbol{r}_k||_2 < \max\left(\text{rtol} * \min\left(||\boldsymbol{b}_k||_2, ||\boldsymbol{r}_0||_2\right), \text{atol}\right)$.

The iterative solver will also stop after a maximum number of iterations (maxits), if neither rtol nor atol are reached.

## 2.2 Software design

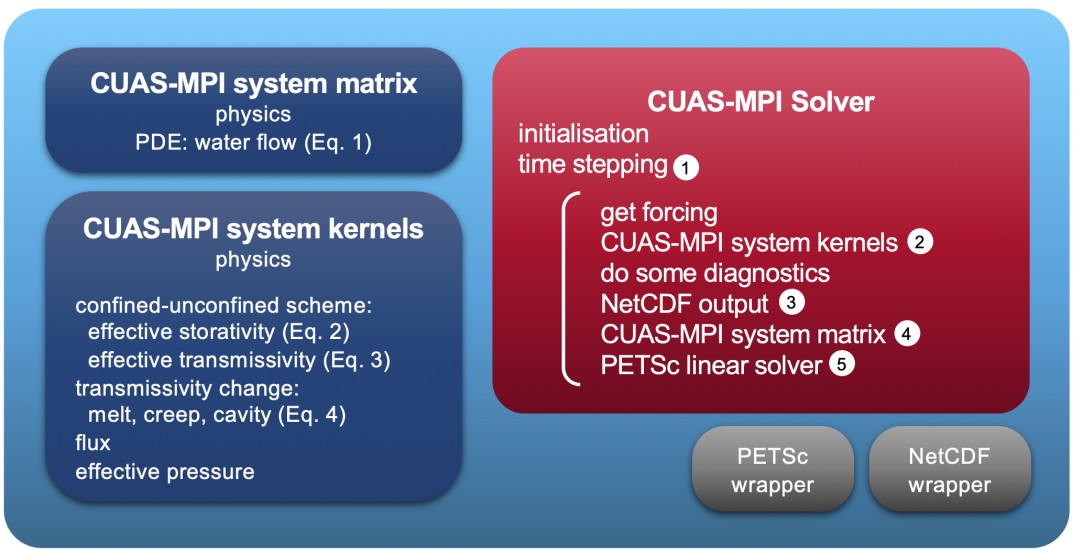

**Figure 2.** Components of CUAS-MPI. The physics modules of the mathematical model are shown in dark blue. The actual solver is sketched in the red box. Grey boxes denote wrappers interfacing to PETSc and NetCDF. The numbers 1–5 in white circles are used again in Section 4, e.g. Figure 8, for cross-referencing.

The starting point of this project was a serial implementation of CUAS (Beyer et al., 2018), partially written in Python. While producing good numerical results, its performance was too low for larger setups such as for the subglacial hydrology under the whole Greenland Ice Sheet. Our new software design for CUAS-MPI is based on this earlier implementation: CUAS-MPI

again uses regular two-dimensional grids and the physics kernels are implemented analogously to the Python implementation. But we designed the data structures and computation of CUAS-MPI to run in parallel on large HPC systems. Each kernel

represents an individual equation of CUAS (see Figure 2 and corresponding equations in Section 2.1). While a kernel reads and writes data from and to grids, the CUAS-MPI system matrix creates the equation system (matrix & vector) based on the previously computed grids. This system of linear equations is then solved by PETSc linear solver. Finally the solution is stored in a grid and the time stepping sequence starts over.

The order of operations in each time step (red box in Figure 2) may seem odd at first, but under normal circumstances, the program would stop after writing the NetCDF output (3) as usual if no more time steps are required. There are two main reasons for the chosen loop structure: First, the model has only two state variables ($h$, $T$), and all derived quantities can be computed using the methods implemented in CUAS-MPI system kernels (lower blue box in Figure 2). This is called "do some diagnostics" in the figure. We want the model output to be always consistent with the state variables without recomputing them prior to the output (e.g. $\nabla h$ is needed for Eq. 4 and later for computing the water flux in Eq. 1). Second, the model can run with minimal output (option "small" in Table 2) for many time steps. If the user later decides that additional output fields are required, a time slice from the model output can be selected using command line tools for manipulating the NetCDF file, and CUAS-MPI can be restarted from that slice configured with a zero time step length to compute only the requested derived quantities (see, e.g. option "normal" in Table 2).

The goal of our code development was to arrive at a software artefact that is performant on current HPC systems but also maintainable in the light of future HPC system evolution. To that end, we based our development on the well-known PETSc (Balay et al., 2021) parallel math library. PETSc is an open source software supported by a wide user base and, for example, part of the software stack supported by the U.S. exascale project https://www.exascaleproject.org/research-project/petsc-tao/. Similarly, to parallelize the NetCDF output, we used the HDF5 library, an open-source product maintained by the not-for-profit HDF group. By basing our development work on these software infrastructures, we profit both from their maturity and performance on current systems, but also, in particular, from performance improvements that will be implemented by the community supporting these software libraries down the road.

In CUAS-MPI, we use PETSc with an object-oriented interface we designed to handle our matrices, vectors and grids. In particular, we employ the distributed array (DMDA) features of PETSc for grid creation, ghost cell update, and optimal matrix and vector distribution in the context of two-dimensional structured grids. The distribution pattern is generated by `DMDACreate`. Grids and vectors are directly derived from this pattern and the matrix is distributed and initialized accordingly, using `DMCreateMatrix`. This ensures that all data structures use the same distribution, which lowers communication cost during assembly operations and the linear solver. We implemented our own grid class, which is based on PETSc DMDA vectors and provides some sanity features, e.g. the grid access is guarded by read and write handles which automatically trigger ghost cell updates between the different parallel processes after each kernel execution. Thus, we are guaranteed not to miss necessary ghost cell updates. The solver can be selected by the user from the list of available direct and iterative PETSc solvers. We use the iterative GMRES solver for the Greenland simulations. For smaller-scale tests, we also employed the direct solver MUMPS (Amestoy et al. (2001), Amestoy et al. (2019)).

The regular two-dimensional grids of CUAS-MPI are stored in the standardized file format NetCDF for in- and output. Different libraries implement parallel read and write operations for NetCDF-files: The NetCDF implementation (Rew and

Davis (1990)) with HDF5, PnetCDF (Latham et al. (2003)) and ParallelIO (Hartnett and Edwards (2021)). As all three parallel NetCDF libraries have advantages and disadvantages, we implement an adapter class which provides an uniform interface of the features we require in CUAS-MPI. Thereby we ensure the flexibility to switch between different I/O implementations and allow easy adaption to future developments. The abstraction layer is able to handle scalars, vectors and grids according to the format used in CUAS-MPI and pass it to the selected NetCDF implementation. The NetCDF library is always used for reading, and we also employ it in our experiments. CUAS-MPI supports four levels of output: small, normal, large and xlarge (Table 2). The small output includes only the absolutely necessary fields. All other fields can be derived. In the three more complex output configurations we write additional analytical information, which is computed in the CUAS-MPI solver pipeline.

## 3 Validation of CUAS-MPI using analytical solutions

Validating the results of CUAS-MPI is an essential task to quantify the computational errors which arise while solving the continuous differential equations in a computationally discrete environment. Specifically, we check the consistency of the numerical solutions of CUAS-MPI with suitable analytical solutions. By doing so, we verify the implementation of the governing equation (Eq. 1) in the confined and unconfined case, as well as the implementation of Neumann and Dirichlet boundary conditions, and quantify their error range. The analytical solutions on which our verification is based were developed in the field of hydrology and are commonly known as pumping tests. The problem we simulate in a pumping test involves a horizontal aquifer of uniform thickness $b$, constant conductivity $K$ and a pump of constant rate $Q^*$ that is located at the centre of the domain and that fully penetrates the aquifer. We compare the numerical results against the analytical solutions in two cases. One in which the aquifer is confined and the model is linear, and another where the aquifer is unconfined and the model is nonlinear. Both the confined and unconfined CUAS-MPI simulations were performed with the direct (MUMPS) and the iterative (GMRES) solver, and the results of the two computational methods were indistinguishable.

### 3.1 Confined case

For the simulation in a confined case the transmissivity is constant and the model (Eq. 1) is linear. Assuming that the aquifer has an infinite extent, the model has the analytical solution (Theis, 1935)

$$s_{\mathrm{ub}}(r_{\mathrm{pm}}) = \frac{Q^*}{4\pi T} W\left(\frac{r_{\mathrm{pm}}^2 S}{4Tt}\right), \tag{5}$$

where $s \equiv h(x,y,0) - h(x,y,t)$ is the drawdown, $r_{\mathrm{pm}}$ is the distance between the pump to the measurement positions, and $W$ is the dimensionless well function. This exact solution can be tested despite the bounded numerical domain, as long as the flow remains far from the boundaries. Indeed, the CUAS-MPI results are in good agreement with the analytical prediction $s_{\mathrm{ub}}$ (Eq. 5) until $\approx 3 \times 10^4$ s, when the flow begins to be affected by the finite boundaries of the numerical domain (Figure 3).

Moreover, an analytical solution that accounts for a bounded domain can be constructed based on the unbounded solution (Eq. 5) using the method of images, which can be applied due to the linearity of the model in this case (Ferris et al., 1962). Specifically, we consider the case where a pump is at equal distance from two Dirichlet boundaries (zero drawdown) and two

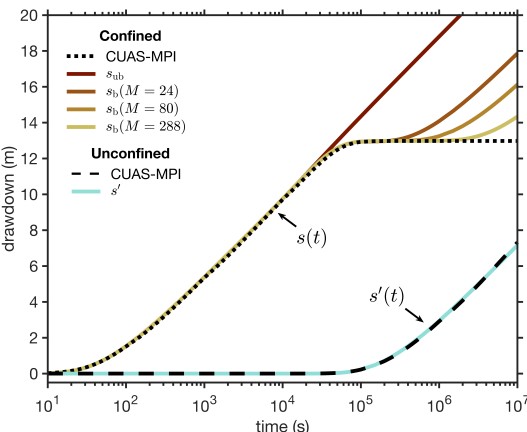

**Figure 3.** Comparison of the CUAS-MPI results for the drawdown to the analytical solutions, for both a confined and unconfined aquifers. In the confined case the results are compared with analytical solutions for an unbounded domain (Eq. 5) and with the analytical solutions for a bounded domain (Eq. 6) with a growing number of image wells ($M = 24, 80,$ and $288$). In the unconfined case, the results are compared with the approximated solution (Eq. 7). The point of drawdown measurement is $80\,\text{m}$ away from the pumping well, for both simulations.

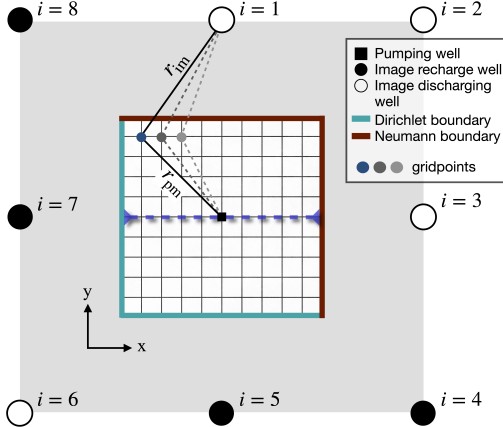

**Figure 4.** Configuration for numerical validation using analytical solutions (pumping test), with Dirichlet conditions at the southern and western numerical boundaries (blue) and Neumann conditions at the northern and eastern numerical boundaries (brown). The wells outside of the numerical domain ($\circ, \bullet$) are the image wells, each described by the unbounded analytical solution (Eq. 5), which collectively construct the solution (Eq. 6) for the bounded domain (Ferris et al., 1962). The blue dashed line indicates the profile of Figure 5.

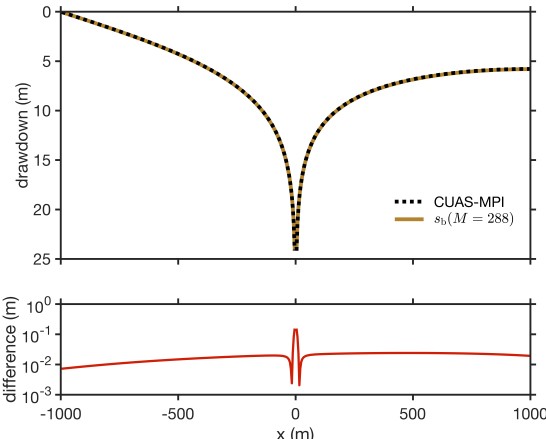

**Figure 5.** (Top) The drawdown along the blue dashed line in Figure 4 computed by CUAS-MPI and compared with the prediction $s_\mathrm{b}$ (Eq. 6), which accounts for the Dirichlet and Neumann boundary conditions on the left and right, respectively. (Bottom) The difference between the computed and predicted drawdown, shown in the top panel, indicates an overall small discrepancy between the numerical and theoretical values.

Neumann boundaries (zero drawdown gradient) (Figure 4), which are the two types of boundary conditions implemented in CUAS-MPI. The analytical solution for such a configuration consists of a superposition of well solutions to image wells placed

across the domain boundaries, as illustrated in Figure 4 (Ferris et al., 1962). Therefore, the analytical solution for the drawdown in a bounded domain $s_\mathrm{b}$ is the series

$$s_\mathrm{b} = s_\mathrm{ub}(r_\mathrm{pm}) + \sum_{i=1}^{M} s_\mathrm{ub}(r_\mathrm{im}), \qquad (6)$$

where $r_\mathrm{im}$ is the distance from the $i$th image well to the point of measurement. The accuracy of the solution grows with the number of image wells $M$.

The numerical simulation for the confined case is set up with a domain size of $2000\,\mathrm{m} \times 2000\,\mathrm{m}$, $b = 100\,\mathrm{m}$, $S_\mathrm{s} = 10^{-6}\,\mathrm{m}^{-1}$, $S = S_\mathrm{s}\,b$, $K = 4.16 \times 10^{-5}\,\mathrm{m\,s}^{-1}$ and an initial hydraulic head of $h(x, y, t = 0) = 300\,\mathrm{m}$. The pumping well has a constant rate of $Q^* = 0.1\,\mathrm{m}^3\,\mathrm{s}^{-1}$. We find that the CUAS-MPI results are in good agreement with the bounded analytical solution $s_b$ (Eq. 6) for a period that grows longer with the number $M$ of image wells (Figure 3, 5).

## 3.2   Unconfined case

For the simulation in an unconfined aquifer the transmissivity is proportional to the head and consequently the model (Eq. 1) is nonlinear. To validate the results of CUAS-MPI in this case, we approximate the drawdown $s'$ of the unconfined aquifer using the drawdown of an equivalent confined aquifer $s_{ub}$ (Eq. 5) with $T = Kh(t = 0)$ through the relation

$$s' = b - \sqrt{b\left(b - 2\,s_\mathrm{ub}(r_\mathrm{pm})\right)}, \qquad (7)$$

which is based on the Dupuit-Forchheimer assumption that the horizontal flux is greater than the vertical flux, and provides a good prediction of the drawdown at large distances from the pump compared to the aquifer thickness (Jacob, 1963).

For this unconfined case, the domain size is set to $4000\,\mathrm{m} \times 4000\,\mathrm{m}$ and the initial hydraulic head is $h(x, y, t = 0) = 99\,\mathrm{m}$. We set the specific yield to $S_y = 0.4$, which represents a subglacial hydrology system with an EPM approach (de Fleurian et al., 2014), and set the pumping well with a constant rate of $Q^* = 0.1\,\mathrm{m}^3\,\mathrm{s}^{-1}$. We find that the simulation results agree well with the approximated analytical solution for the unconfined case (Figure 3).

## 4 Performance of CUAS-MPI running a representative Greenland setup

CUAS-MPI was developed to enable high-performance and high-throughput simulations on up-to-date HPC systems, which typically consist of many-core nodes connected by a high-speed network. To assess the performance of CUAS-MPI, we present performance data of CUAS-MPI on the Greenland Setup described in Section 4.1. We run the model for 24 time steps (one model day) using different grid resolutions to compare the model performance and scaling behavior. We use the GMRES linear solver with a Jacobi preconditioner.

We employ the compiler GCC 11.2, the message passing library OpenMPI 4.1.4, the math library PETSc 3.17.4 and the data format libraries NetCDF 4.7.4 and HDF5 1.8.22 for our performance experiments. To instrument the code for measurements, we use Score-P 7.1. Score-P is a highly scalable performance measurement infrastructure for profiling and event tracing. The code is compiled using optimization level "-O3" and native processor architecture flags "-march=cascadelake". All experiments are conducted on dedicated compute nodes of the Lichtenberg HPC system (https://www.hrz.tu-darmstadt.de/hlr/hochleistungsrechnen/index.en.jsp) with two 48-core Intel Xeon Platinum 9242 processors and 384 GB of main memory each, connected with an InfiniBand HDR100 fat tree network providing point-to-point connections between nodes. The Slurm scheduling system is used for workload management. As such, the Lichtenberg HPC system is representative of the currently most commonly used type of HPC system. Due to temporary energy saving measures, turbo-mode has been disabled and the base frequency of the chips reduced by 100 Mhz to 2.2 GHz. Turbo-mode allows the CPU to exceed its base frequency for short time, but also considerably increases power consumption. Every experiment is repeated three times, and the average runtime is reported. For all runs of the models with 1200 m or finer resolution, we see a relative standard deviation of the time of less than 5%. For the coarsest 2400 m model, we observed a relative standard deviation of the time of 15 %, which we believe to be in part due to the very short running time of the code in this case.

### 4.1 Description of the Greenland setup

The model domain consists of a rectangular area containing the Greenland ice sheet. Grid points of the hydrological system for which Eq. 1 is solved are colored red in Figure 6. Boundary conditions are determined by the type of grid cell next to the margin grid cells of the subglacial system. In case of a land terminating margin, we use a zero-flux Neumann boundary condition (no-flow). A transition to the ocean at the grounding line is a Dirichlet boundary condition for the head. We use $h = 0$, at the grounding line assuming the water pressure in the aquifer equals the the hydrostatic ocean pressure. We ignore the

slight density difference between sea water (about $1028\,\mathrm{kg\,m^{-3}}$) and fresh water ($1000\,\mathrm{kg\,m^{-3}}$). By our choice of boundary conditions, the subglacial water is drained into fjords only, as land terminating margins are prohibiting outflow. This is not very realistic, as subglacial water could also feed into ice marginal lakes and rivers and would decrease the hydraulic head (water pressure) in the subglacial system. Outflow could be indirectly simulated by applying Dirichtlet conditions for the hydraulic head at those river and ice-marginal lake locations. The model is capable of doing so, but to our knowledge, no data set of the river and ice-marginal lake locations and fluxes for the Greenland Ice Sheet exists. Even though outflow along the land terminating margin is not considered, we think the setup is realistic enough to draw conclusions about the model performance.

The type of grid cell (mask), the ice thickness, the bedrock and surface topographies are based on a preliminary version of the BedMachine Greenland dataset as used in Christmann et al. (2021) that later became BedMachine version 4 (Morlighem, 2021; Morlighem et al., 2017). The BedMachine dataset is originally available in 150 m resolution (G150), which we regrid to 300 m (G300), 600 (G600), 1200 m (G1200), and 2400 m (G2400) resolution using conservative remapping for the geometry and near-neighbor interpolation for the mask. Ice sheet basal melt from the Parallel Ice Sheet Model (PISM) output (Aschwanden et al., 2016, 1200 m resolution) is regridded to the CUAS meshes using bi-linear interpolation. Large areas consisting of small glaciers get a spatially uniform ice thickness of 1 m assigned in the BedMachine dataset due to insufficient data coverage. This is particularly visible along the south-eastern and eastern margin of the ice sheet. Those areas are also not well resolved in the PISM model. Therefore a minimum ice thickness constraint of 10 m is applied to eliminate thin marginal areas. The observed surface velocity (MEaSUREs, Joughin et al., 2016, 2018) is used to check if areas with at least $30\,\mathrm{m\,a^{-1}}$ are included in the mask. If not, the minimum ice thickness constraint is applied and the bed elevation is adjusted to be consistent with the surface elevation from BedMachine. Further, a flood-filling algorithm (van der Walt et al., 2014) is used to select only the connected grid points from the main ice sheets without peripheral ice caps and glaciers to ensure consistent coverage from BedMachine and the PISM model output. This step is important, because missing data in the basal melt forcing would degrade the solver performance in CUAS-MPI.

The resulting numbers of total and active grid points are given in Table 3. Water input is the basal melt rate presented in Aschwanden et al. (2016). The model parameters (see Table 1) are mainly taken from Beyer et al. (2018) with only two exceptions. We use $S_\mathrm{y} = 10^{-6}$ instead of 0.4, and a minimum transmissivity bound $T_\mathrm{min}$ of $10^{-8}$ instead of $10^{-7}\,\mathrm{m^2\,s^{-1}}$. Those changes are found to result in smoother hydraulic head in areas of no or only very little basal water supply. For the purpose of this study, the exact representation of the Greenlandic hydrological system is not of primary importance, as we analyze the performance of the model, but we represent the Greenlandic Ice Sheet sufficiently realistically to be able to infer from the outcome in this study the computational costs for other simulations. The hydraulic head is initialized to be equal to the ice overburden pressure at each grid point and the initial transmissivity is spatially uniform ($T(t=0) = 0.2\,\mathrm{m^2\,s^{-1}}$). The convergence criteria for the iterative GMRES solver are configured as $\mathrm{rtol} = 10^{-7}$ (relative norm) and $\mathrm{atol} = 10^{-5}$ (absolute norm) with a maximum number of iterations set to $10^5$.

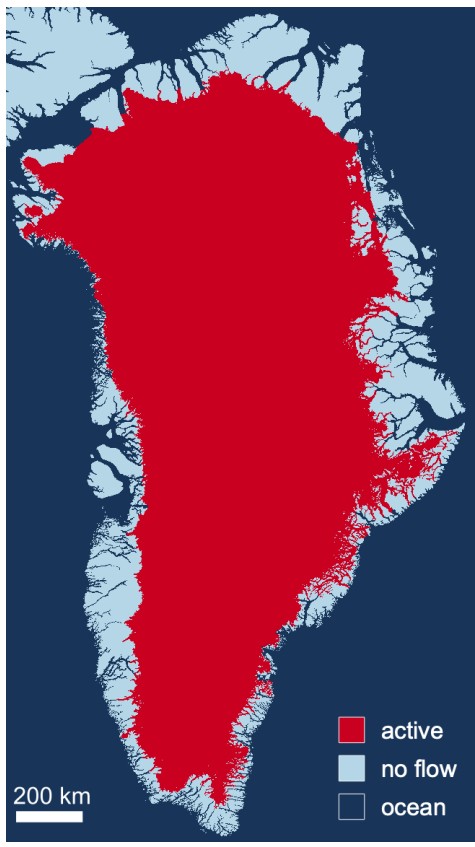

**Figure 6.** The Greenland setup used for this study. Red color represents the area where the hydrological system is computed, dark blue denotes ocean and pale blue land area. Ocean and land lead to different boundary conditions.

## 4.2 Thread occupancy of compute nodes

The two processors on one node share access paths to main memory and they have a sophisticated cache architecture. The sparse matrix setup employed in the numerical solvers requires a significant amount of memory accesses and, when employing all cores, will more than saturate the available memory bandwidth. Thus, it is worthwhile to explore whether a lower thread occupancy on a node, which provides more individual bandwidth to the remaining threads, is not, in the end, the better choice. To this end, we tested CUAS-MPI on the Greenland setup with G600 resolution using full, half, quarter and one-eighth populated compute nodes of the Lichtenberg HPC system. Each thread is pinned to one CPU core and realizes one MPI process. Hence, in our discussion and for our setup, the notions of thread and MPI process are used interchangeably.

The result is shown in Figure 7. Here, the color of a circle indicates the thread occupancy ratio, from one-eight, i.e. 12 threads of a 96-core node, in green, up to the use of all 96 threads on a node, in blue. The size of the circle indicates the hardware investment, i.e. the resources requested in the Slurm script. The smallest circle indicates that only one node was used, the next

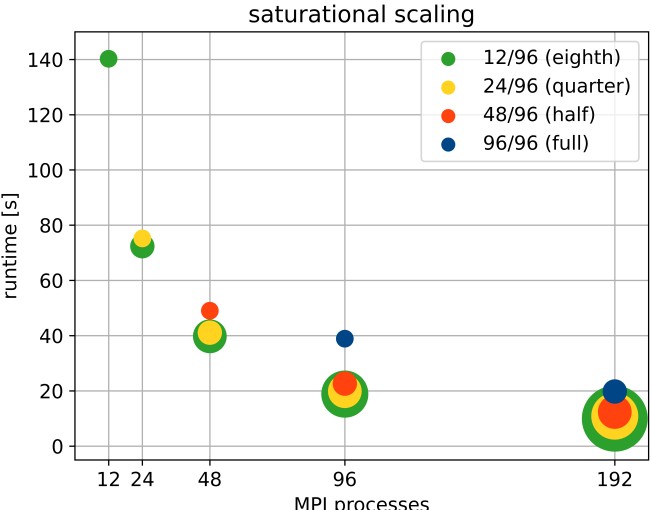

**Figure 7.** Performance of CUAS-MPI on full (96/96), half (48/96), quarter (24/96), eighth (12/96) populated nodes of Lichtenberg HPC system without file output. The center of the circle shows the number of spawned MPI processes (x-axis) and the runtime (y-axis). For each color the smallest circle indicates the usage of one compute node, the next size up indicates two nodes, the third four nodes, then 8 and lastly 16 nodes.

size up indicates two nodes, then four nodes, and lastly 16 nodes. The center of the respective circle indicates the runtime that

was achieved with this setup.

Hence, the leftmost green circle (12 threads on one node, resulting in a wall-clock time of about 140 seconds) is as large as the blue circle in the middle (96 threads one one node, resulting in a runtime of about 40 seconds). On the other hand, the rightmost blue circle is bigger, because here we need to employ two nodes at full occupancy to realize the 192 threads, resulting in a runtime of about 20 seconds. Almost concentric circles, such as the red, yellow and green circles for 192 processes, then

indicate that the additional hardware investment does not pay off, as the corresponding runtime can be achieved with the setup corresponding to the smallest circle.

We see that there is not much difference in the runtime of one node with 48 MPI processes (red circle at 48 processes) to one node with 96 MPI processes (blue circle at 96 processes), but 96 MPI processes on two nodes (red circle at 96 processes) are about twice as fast. The rightmost circles show that 192 MPI processes are faster than 96 MPI processes and that we should

use a distribution of four nodes (red circle), because it is faster than two nodes (blue circle) and not significant slower than four (yellow circle) or eight (green circle) nodes. Similar observations can be made for 96 MPI processes, where an occupancy lower than a half results in some, but not very significant, speedup. Hence, we consider 48 MPI processes per node, i.e. half thread occupancy of a node, as a good trade-off between getting the solution fast and using the hardware reasonably and employ 48 MPI processes on each node to analyze the throughput, i.e. how many simulated years we can run in a day of compute time

(simulated years per day, SYPD), and scalability of CUAS-MPI in the following subsections.

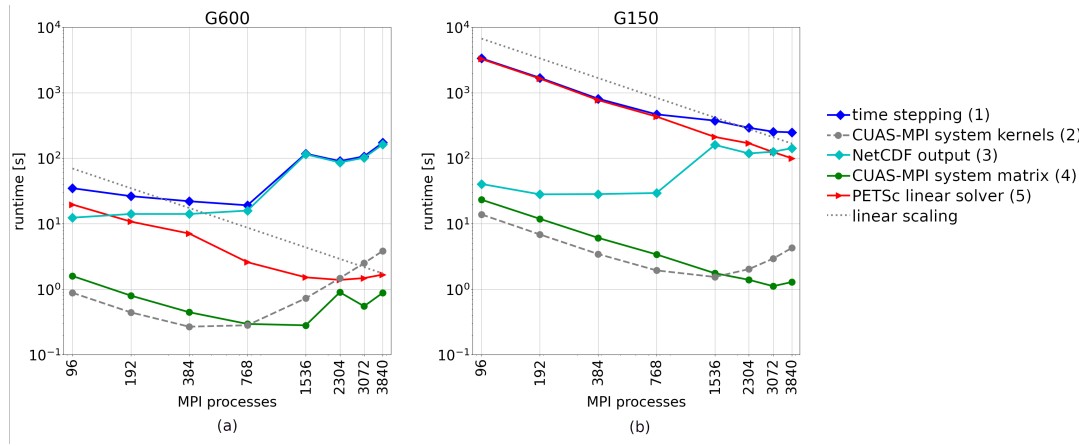

**Figure 8.** Runtime of 24 time steps of CUAS-MPI pipeline writing a single output running G600 (a) and G150 (b).

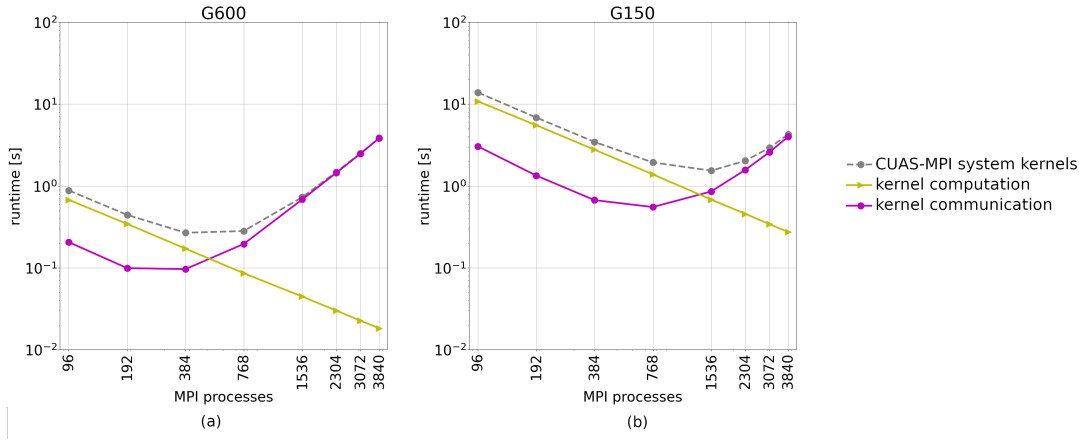

**Figure 9.** Sum of the CUAS-MPI system kernels runtime of 24 time steps running G600 (a) and G150 (b).

### 4.3 Runtime and throughput measurements

We identified five functional parts in CUAS-MPI, whose individual performances are worth differentiating: CUAS-MPI setup, CUAS-MPI system kernels, CUAS-MPI system matrix, PETSc linear solver, and NetCDF output. CUAS-MPI setup contains initial model loading from disc and the setup of necessary grids. The runtime of this code component is not significant in large productive runs and we do not consider it further. The four remaining functional parts are identified in Figure 2. In particular, the CUAS-MPI system kernels category includes all kernels running on CUAS-MPI grids (see the blue box in Figure 2). The grids are necessary for the computation of the system matrix entries and for diagnostic purposes. CUAS-MPI system matrix denotes the creation of the equation system, i.e. the computation of the matrix entries and the matrix assembly, that is solved in the PETSc linear solver. PETSc linear solver is a library call to the solver, in our case the preconditioned GMRES implementation.

Finally, NetCDF output category includes all calls which write data to file during the simulation. The file output performance, in general, greatly depends on the storage and network capabilities of the current HPC system and on the output frequency, which depends on the particular experimental setup in question.

In Figure 8 we show the runtime of different code categories for 24 time steps of CUAS-MPI with one output to disc. The top line of the runtime plots shows the runtime of the entire time stepping sequences, i.e. the aggregate runtime of all other

routines listed as well as forcing calculations and diagnostics, whose runtime is negligeable compared to the code categories shown. First, we note that NetCDF output routine does not scale at all. Its runtime is, for both G600 and G150, essentially flat up to 768 MPI processes, and then increases by an order of magnitude. The general file output performance is highly system and load dependent, as shown by Orgogozo et al. (2014). Therefore we do not investigate this issue further, but keep it in mind for experiment planning.

Considering the coarser G600 grid, we note that, ignoring NetCDF output, the PETSc linear solver dominates runtime and scales up quasi linearly up to 1536 MPI processes, reaching a minimum at 2304 MPI processes. Finally, it is overtaken by the CUAS-MPI system kernels, whose runtime increases past 768 MPI processes, due to the increasing communication overhead between MPI processes and decreasing computation performed on each MPI process.

For the finer grid (G150), on the other hand, we see a continual decreases in runtime as we increase the number of MPI

processes. Ignoring NetCDF output, the PETSc linear solver dominates the runtime, but scales in an almost linear fashion up to 3840 MPI processes. In particular, we see approximately linear scaling also for the CUAS-MPI system kernels up to 768 MPI processes and the system matrix routine up to 3072 MPI processes. Further scaling increases the runtime of these two parts, due to communication overhead.

Disregarding file output operations, the CUAS-MPI system kernels category is the first component reaching its scaling

limit. As all kernels behave in the same way, we consider them as a group. Figure 9 shows the accumulated runtime and the separated runtime of kernel computation and kernel communication for grid data exchange caused by PETSc. We notice, that the computation scales linearly as expected, but the grid exchange prevents further scaling.

In our studies of throughput, i.e. how many simulated years we can run in a day of compute time (simulated years per day, SYPD), we employ the same time step (1 hour) for the different resolutions in order to allow a sensible comparison of the

various code components across resolutions and process counts. This is conservative for the lower resolutions, as generally coarser grids allow larger time steps, because time steps are adapted to fit the stability conditions of the simulation. The different grid resolutions require different numbers of iterations for convergence.

We simulate 1 day (24 time steps) and write one output per day using the large configuration (see Table 2) and measure the runtime. Based on that measurement we compute the runtime for 365 days and compute SYPD.

In Figure 10, we compare the throughput and runtime scaling of different grid resolutions on the Greenland setup, from 96 up to 3840 MPI processes and for different output frequency. Figure 10a intends to provide users the databasis for their simulation planning. As an example, it allows to assess how more/less costly a higher/lower spatial resolution is, if the output frequency is fixed. We have chosen daily and annual output, to cover the upper and lower bounds in user requirements.

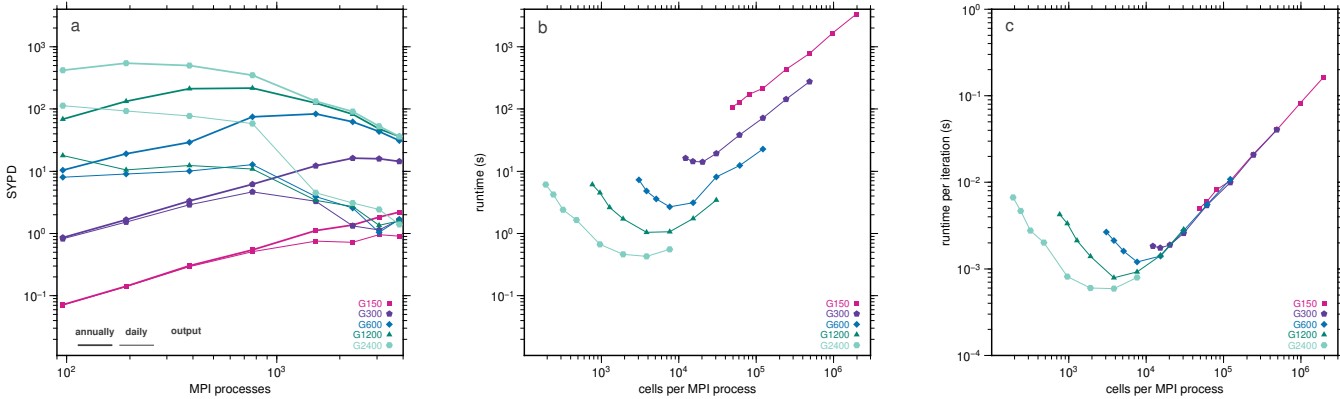

**Figure 10.** Throughput (panel a) versus MPI process, runtime (panel b) and runtime per linear solver iteration (panel c) versus number of cells per MPI process.

The peak throughput of the coarsest grid G2400 of about 550 simulated years per day is reached at 192 MPI processes and annual writing, while the finest grid G150 can efficiently utilize more than 3840 MPI processes for both daily and annual writing. For G150, we derive a maximum throughput of approximately one simulated years per day if daily output is written and two simulated years per day if output is written annually.

In general, there is no sense in using a large number of processes for coarse grids, as there is not enough work to be done for an efficient parallelization. On the other hand, for the G150 grid (which has almost 100 times as many grid points as the G1200 resolution, see Table 3), it does make sense to use more processes to increase throughput: With 768 and 1536 processes, we can compute 198 and 402 model days per compute day. So, in particular, going from 768 to 1536 processes, we increase throughput by a factor of two.

The fact that there is no sense in using many processes for coarse grids is underscored by panel b of Figure 10, which shows the runtime of the CUAS-MPI pipeline without output against the number of cells per MPI processes. For any given resolution, the leftmost data point corresponds to the run with 3840 processes, i.e. the maximum number of processes employed in our study, whereas the rightmost data point corresponds to the run with 96 processes, i.e. the minimum number of processes employed in our study. The line for the coarsest grid G2400 is the leftmost one, and we see that here, using more than 192 processes (the inflection point of the curve) makes no sense as we simply exercise communication overhead and do not have enough work on each process to offset this overhead. The lines for finer resolutions then progressively shift right, and at the finest resolution G150 we do not have an inflection point any more - even for 3840 processes there is enough work to be done locally. So, in particular the finest grid G150 still has potential to effectively use even more processes.

The number of iterations taken by the iterative linear solver varies with grid resolution, and more iterations are needed for solving the linear equation system for finer grids. While this is certainly an issue to consider for throughput, for scalability, i.e., the question of whether using more processes reduces computation time, the time of one iteration is the deciding factor. Hence, we also show in Fig 10c the runtime per linear solver iteration versus the number of cells per MPI process. The minimum of

the curves does not change, but the curves move closer together, as most of the compute time is dependent on the number of cells per MPI process. However, the minimum of the total runtime per linear solver iteration is increasing as the resolution gets finer, because we use more MPI processes at the respective minima and more MPI processes cause more communication overhead than less MPI processes, e.g. for global communications for norm computations.

Figure 10 has also practical use for planning simulations. For most tasks, scientists have a constraint on SYPD for a simulation, such as simulations must not take longer than a certain amount of hours/days. The research question we want to answer also determines the output frequency, e.g. investations of seasonality need daily output. Using Figure 10 one can also, e.g., estimate the affordable spatial resolution and select the optimal number of MPI cores, or estimate compute time for proposals for compute resources.

## 5 Discussion

The throughput shown by CUAS-MPI enables ice sheet wide simulations in high (600 m), but potentially not highest resolution (150 m BedMachine grid resolution) due to the computational costs. The number of time steps required for a seasonal cycle limits the number of years that can be simulated within a few days compute time as can be seen in Figure 10. Derived from the timings we measure on the Lichtenberg HPC system, a simulation covering the 90 years from 2010 to 2100 in 600 m spatial and one hour temporal resolution requires an estimated wall-clock time of 74 hours (3 days) on 384 MPI processes. Such a simulation can be performed a few times, but is not feasible for ensemble simulations for different atmospheric inputs. Moreover, a high computational demand arises from spin-ups for having a proper initial state for simulations and a control run needed for assessment of the projection run. So, with CUAS-MPI, Greenland-wide simulations are still challenging, but they are feasible. The computational costs are also emphasising the need for efficient coupling of ice-sheet and hydrology models.

Although we have applied the code to an entire ice sheet, applications to alpine glaciers might be of interest for a larger community. As an example we consider the Kanderfirn Glacier (Switzerland) which has a size of about $12\,\text{km}^2$. To compute this glacier in the $10\,\text{m}$ resolution would result in 120.000 grid points. Assuming hourly time steps and daily output, one year would require about 3 minutes wall-clock time on 384 MPI processes (based on G150 performance).

The physical (and mathematical) model of CUAS-MPI may, of course, not be powerful enough to simulate the complex physics sufficiently well in other cases of interest. However, with CUAS-MPI domain scientists now have a tool to conduct simulations on relevant areas of interest to identify strengths and weaknesses of the physical basis of the model. The code has been designed in a modular fashion to allow for extensions of the underlying physical model.

The next step will be the coupling of CUAS-MPI to an ice sheet model, in our case the finite element based Ice-Sheet and Sea-Level System Model (ISSM, https://issm.jpl.nasa.gov/, Larour et al. (2012)). There are inherent system dependencies between the physical quantities in both models, ice sheet and hydrology, hence a coupling needs to consider the ingestion of a larger number of fields from the ice sheet code into CUAS-MPI, while there is only the effective normal pressure to be fed into the ice sheet model. To this end, we plan to employ the preCICE coupling library for partitioned multi-physics simulations (Chourdakis et al., 2022).

As an alternative to coupled simulations, one can also consider the possibility to implement the subglacial hydrology model of CUAS-MPI directly in ice sheet codes like ISSM. Such a monolithic implementation would avoid the necessity of inter-model communication. However, we see various advantages in coupling the two codes versus a monolithic solution. First, a standalone implementation supporting a generalized coupling interface is more flexible and can be used in many other projects. Second, it is fully independent of other codes discretization and thus prevents inconsistencies. Finally, we see a huge advantage in the implicit additional parallelization potential. While a monolithic implementation most likely will execute the ice and hydrology modules one after the other in a multi-physics environment, a coupled implementation can easily run ice sheet and subglacial hydrology simulation on dedicated nodes. So both modules can run in parallel and exchange data after computing the next time sequence. Certainly the data exchange of coupled simulations generates additional overhead, but as we see no necessity for high frequency data exchange, e.g. each time step, but consider less frequent exchange intervals, e.g. daily or even seasonal, the overhead of coupled simulations will be affordable and the benefit of running hydrology in parallel to other modules will prevail.

For this endeavor, it is worth comparing the computational costs of the ice sheet and hydrology models. Comparing the SYPD for a Greenland setup in the ice sheet model in the highest tested resolution (minimum edge length of 250 m) presented in (Fischler et al., 2022) with the finest grid of the CUAS-MPI setup (edge length of 150 m), we find that the computational costs are comparable for both models. As a consequence, both simulations would need about the same time per year, thus achieving low idle time at synchronization points in coupled runs. Nevertheless, the coupling frequency depends on the phenomena investigated in both models. If someone wants to study the effect of ocean tides on the subglacial environment, time steps in the hydrology model need to resolve the tides, and the ice sheet model may or may not be informed about the changes depending on the scientific goals and ice sheet model capabilities.

Considering the performance of CUAS-MPI, we see that the NetCDF performance plays an important role and limits scalability of CUAS-MPI on the Lichtenberg HPC system system that we are running on. Hence, we suggest to use low-frequency output, e.g. annual output. In addition, the smallest output configuration can be used to reduce the output costs, and then the CUAS-MPI restart capabilities can be used to compute additional fields afterwards. We have not investigated NetCDF performance further, as I/O performance is highly system dependent. We have, however, encapsulated I/O in our software design so that other NetCDF implementations can be linked in easily.

Outside of I/O, the runtime of CUAS-MPI is dominated by the runtime of the PETSc linear solver (unless the local work becomes too little). Our scaling tests have shown that the solver still has more scaling potential in the case of large setups, but we reached the scaling limit of coarser grids. Future throughput improvements will be enabled by improvements of the PETSc software infrastructure, which is being continually improved and updated, e.g. using persistent MPI communication.

The scaling of the CUAS-MPI system kernels, which generally is of lower importance overall, might be further improved by asynchronous communication or additional thread parallelism per MPI process to increase computational granularity, i.e. the amount of work which is performed by a task. Hybrid parallelism would also enable the opportunity to use less parallelism in less scalable regions like file output and more parallelism in the computationally dominant linear solver.

## 6 Conclusions

CUAS-MPI is a newly designed process-parallel code, software-engineered for performance and portability, based on the model of Beyer et al. (2018), which so far was implemented as a prototype. The results of CUAS-MPI are consistent with analytical solutions of pumping tests in hydrology, implying successful code implementations of the Beyer et al. (2018) model. The code we present here can be applied widely for glaciological applications, from simulating water discharge of smaller glaciers for hydropower production to estimating seasonality of water flux of rivers that are fed by subglacial water.

The code is instrumented for performance measurements, which we conducted on the Lichtenberg HPC system at TU Darmstadt. Our study demonstrates, that CUAS-MPI performs well and scales up to 3840 MPI processes, enabling the simulation of challenging setups at finer resolutions. Some performance limitations result from the current implementation of PETSc and NetCDF, and we anticipate that CUAS-MPI will profit from performance improvements in these software infrastructures in the future.

The insights gained from the performance measurements can be used to plan simulations. Given a constraint on available compute hours, the numbers presented here allow to assess the choices of spatial resolution and output frequency, as well as to select the optimum number of MPI-cores.

Subglacial hydrology and ice sheet evolution are strongly related to each other, so runtime coupling of CUAS-MPI and ice-sheet simulations like ISSM is an important topic. As the subglacial hydrological system delivers freshwater into fjord systems and ice shelf cavities, also the coupling to ocean models will be needed in future. Therefore we see necessary enhancements in the implementation of inter-simulation data exchange and the adaption of the model to support changing simulation domains in transient runs.

In contrast to the Greenland Ice Sheet, the hydrology below the Antarctic ice sheet is expected to be driven by basal water production without additional water from the ice sheet surface draining to the base during the summer season. Nevertheless, subglacial hydrology seems very important and two-dimensional subglacial hydrology models have already been applied to large Antarctic catchments (Dow et al., 2022; Dow, 2023). The performance of a subglacial hydrology model is vital if model simulations are extended to the entire Antarctic Ice Sheet, which is several times larger than the Greenland Ice Sheet.

Our study demonstrates, that CUAS-MPI fulfills a number of the demands of the Earth System Modelling community: allowing simulations from global to local, performance portability, scalable workflows and being open source.

*Code and data availability.* We published our repository on https://github.com/tudasc/CUAS-MPI. For the measurements presented in this paper, version 0.1.0 was used, which is available at github and through the following address https://doi.org/10.5281/zenodo.7554686. Our profiling data is available on https://doi.org/10.48328/tudatalib-1034.

*Author contributions.* Y.F., C.B., T.K. and A.H. planned the project. Y.F. and C.B. developed the software design. Y.F., L.O., R.E. and T.K. implemented the code. Y.F. conducted all performance measurements. Y.F., C.B., T.K. and A.H. analyzed the performance measurements.

T.K. developed the Greenland setup. T.K. and A.H. run the code for polar applications. J.S. and R.S. developed the pumping test concept, J.S. conducted the tests. All authors discussed the performance analysis and wrote the manuscript.

*Competing interests.* The authors declare that they have no conflict of interest.

*Acknowledgements.* The authors gratefully acknowledge the computing time provided to them on the high-performance computer Lichtenberg 2 at the NHR Center NHR4CES at TU Darmstadt under grant p0020118. NHR4CES is funded by the Federal Ministry of Education and Research, and the state governments participating on the basis of the resolutions of the GWK for national high performance computing at universities.

Parts of this work were funded by the German-Israeli Fund GIF under grant number I-1493-301.8/2019.

The authors thank Vadym Aizinger (University of Bayreuth) and Alexander Hück (TU Darmstadt) for helpful discussions.

## Appendix A: Workflow

We usually prepare a setup script to pre-process available datasets and set up the simulations. In that script, the model domain and the grid are defined and used to create the mask (see Figure 6). This mask is one of the input fields and informs the model about the locations and types of boundaries. In addition, the fields for bedrock topography, ice thickness and water input are prepared into one NetCDF file.

The next step is initialisation. The initial head and thus the initial water pressure can be selected via specific named CUAS-MPI options to be spatially uniform, following the bed elevation or to be equal to the ice overburden pressure. The initial transmissivity can be set to a spatially uniform value via a command line option. Full control over the initial conditions is further given using the CUAS-MPI "restart from file" capabilities. A restart could be done from, e.g. the last time slice of a previous run, from arbitrary fields for head and transmissivity provided by the user or from the output of a coarse resolution spin-up that has to be remapped onto the new grid outside of CUAS-MPI. All file names and parameters used for a model run are stored in the NetCDF output file for later reference.

The last step is to set up the actual run script that serves the needs of the cluster environment. In our case this is the Slurm scheduler. This step includes setting up the command line options to control CUAS-MPI's physics and time stepping, setting up the the PETSc Solver via the environment variable `PETSC_OPTIONS`, as well as the memory, node, core and runtime configuration on the cluster. The time stepping strategy can be provided with an optional parameter specifying either the constant time step or providing the sequence of time steps to be applied in a time step file. Our implementation is backwards compatible to the setup of the serial version, supporting input data in NetCDF format and the same command line parameters. The comprehensive list of the parameters is displayed when executing the CUAS-MPI executable with the "–help" option. In addition, CUAS-MPI is able to restart from previous runs.

A typical use case may be the simulation of a seasonal hydrological cycle. To this end, a spin-up would be conducted first to retrieve a steady-state system. This may be done on a coarse spatial resolution first, followed by a refinement using another grid using the restart option. At this point, the actual simulation with seasonal forcing starts using the refined mesh. It is highly recommended to also run a control simulation on the refined mesh by further using the steady-state forcing. This allows us to account for model transients that may still be contained in the model after the spin-up. If a particular target area needs to be simulated in even higher resolution, a regional subdomain nesting can be employed to reduce the computational costs of a model run. If a drainage basin can only be covered partially, it is very beneficial to embed this area into a large-scale and probably also coarse resolution run that provides boundary conditions along the subdomain margin (Dirichlet conditions for $h$ and $T$). The desired output frequency, as well as the variables to be stored, can be adjusted to the needs of the user depending on the domain and resolution.

Another typical application is a projection of the change of the hydrological system over a larger time period. For this purpose a spin-up is required, too. To balance the costs for the long simulation period, the user can reduce spatial resolution if the science questions allows, or reduce the output to only annually write files with the essential physical fields. In all cases the user needs to choose an adequate time step.

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

**Table 1.** List of symbols and parameters used in the model. The values given in the lower part of the table refer to the Greenland setup.

| Symbol | Name | Value / Units |
|---|---|---|
| $h$ | hydraulic head | m |
| $T$ | transmissivity | $\mathrm{m^2\,s^{-1}}$ |
| $S$ | storage | - |
| $t$ | time | s |
| $z_\mathrm{b}$ | bed elevation | $\mathrm{m^{-1}}$ |
| $Q$ | water supply per unit area | $\mathrm{m\,s^{-1}}$ |
| $Q^*$ | water supply | $\mathrm{m^3\,s^{-1}}$ |
| $\mathbf{q}$ | depth-integrated water flux | $\mathrm{m^2\,s^{-1}}$ |
| $T_\mathrm{e}$ | effective transmissivity | $\mathrm{m^2\,s^{-1}}$ |
| $S_\mathrm{e}$ | effective storage | - |
| $a_\mathrm{melt}$ | opening by melt | $\mathrm{m^2\,s^{-2}}$ |
| $a_\mathrm{cavity}$ | opening by sliding | $\mathrm{m^2\,s^{-2}}$ |
| $a_\mathrm{creep}$ | opening/closure by creep | $\mathrm{m^2\,s^{-2}}$ |
| $p_\mathrm{w}$ | water pressure | Pa |
| $p_\mathrm{i}$ | ice pressure | Pa |
| $N$ | effective pressure | Pa |
| $H$ | ice thickness | m |
| $s$ | drawdown | m |
| $r_\mathrm{pm}$ | distance between the pumping well and measurement positions | m |
| $r_\mathrm{im}$ | distance between an image well and measurement positions | m |
| $T_\mathrm{min}$ | min. transmissivity | $10^{-8}\,\mathrm{m^2\,s^{-1}}$ |
| $T_\mathrm{max}$ | max. transmissivity | $100\,\mathrm{m^2\,s^{-1}}$ |
| $S_\mathrm{s}$ | specific storage | $9.8 \times 10^{-5}\,\mathrm{m^{-1}}$ |
| $S_\mathrm{y}$ | specific yield | $10^{-6}$ |
| $b$ | EPM thickness | $0.1\,\mathrm{m}$ |
| $d$ | confined / unconfined transition | $0\,\mathrm{m}$ |
| $A$ | creep rate factor | $5 \times 10^{-25}\,\mathrm{Pa^{-3}\,s^{-1}}$ |
| $K$ | hydraulic conductivity | $10\,\mathrm{m\,s^{-1}}$ |
| $L$ | latent heat of fusion | $334\,\mathrm{kJ\,kg^{-1}}$ |
| $n$ | flow law exponent | 3 |
| $g$ | gravitational acceleration | $9.81\,\mathrm{m\,s^{-2}}$ |
| $\mathbf{v}_b$ | basal ice velocity | $10^{-6}\,\mathrm{m\,s^{-1}}$ |
| $\beta$ | cavity opening parameter | $5 \times 10^{-4}$ |

| configuration | enabled fields for output | size per time step | |
|---|---|---|---|
| | | G600 | G150 |
| small | head, transmissivity | 188 MB | 3 GB |
| normal | bed elevation, water input, | 752 MB | 12 GB |
| | dT/dt due to channel wall melt & creep opening & cavity opening, | | |
| | effective pressure, flux | | |
| large | ice thickness, effective layer transmissivity & storativity | 940 MB | 15 GB |
| xlarge | ice pressure | 940 MB | 15 GB |

**Table 2.** Output options of CUAS-MPI, the categories are inclusive, each option includes the options above.

| name | resolution | grid points | active (%) |
|---|---|---|---|
| G150 | 150 m | 187 459 428 | 39.08 |
| G300 | 300 m | 46 879 140 | 39.05 |
| G600 | 600 m | 11 726 928 | 39.03 |
| G1200 | 1200 m | 2 935 305 | 39.01 |
| G2400 | 2400 m | 734 720 | 39.06 |

**Table 3.** Characteristics of the Greenland setups, where 'active' is the number of active grid points (red in Figure 6) relative to the total number of grid points.