# Peer review of "A parallel implementation of the confined-unconfined aquifer system model for subglacial hydrology: design, verification, and performance analysis (CUAS-MPI v0.1.0)"

_Geoscientific Model Development, 2022_

## Author Comment (AC1)

**GMD Reviews and Authors' Response concerning the paper "A parallel implementation of the confined-unconfined aquifer system model for subglacial hydrology: design, verification, and performance analysis (CUAS-MPI v0.1.0)"**

Yannic Fischler[1], Thomas Kleiner[2], Christian Bischof[1], Jeremie Schmiedel[4], Roiy Sayag[4], Raban Emunds[1,2], Lennart Frederik Oestreich[1,2], and Angelika Humbert[2,3]

[1]Department of Computer Science, Technical University Darmstadt, Darmstadt, Hesse, Germany
[2]Alfred-Wegener-Institut, Helmholtz-Zentrum für Polar- und Meeresforschung, Bremerhaven, Bremen, Germany
[3]Faculty of Geosciences, University of Bremen, Bremen, Germany
[4]Department of Environmental Physics, BIDR, Ben-Gurion University of the Negev, Sde Boker, Israel

**Correspondence:** Yannic Fischler (yannic.fischler@tu-darmstadt.de)

*Copyright statement.* ©2023 all rights reserved

**1 General Comments**

We thank the anonymous reviewer for the helpful comments on the manuscript. We will extend the entire description of the state of the art of hydrology, the description of the hydrology of this model and the description of the modeling of the pumping tests. In return we moved Section 2.3 into appendix. We will incorporate the vast majority of the minor suggestions.

**2 Review 1 by an anonymous referee**

The authors present a novel HPC modelling tool for subglacial water fluxes, for which they produce a validation and a computational performance assesment on a case set up at the whole Greenland scale. Finally they discuss its perspectives of applications.

The paper is clearly organised and it includes a careful study of the parallel behaviour of the developed modelling tool, with a module-wize quantification of the scalabilty. This point of especially important due to the limitations of applicability of modelling strategies that are related to too long computation times. The potential fallouts of this work are important, and adequatelly discussed. Nevertheless the manuscript suffers from formal flaws and is sometimes too elliptical – see the general and specific comments below.

So I suggest that this manuscript should be accepted for publication in GMD after minor revisions for applying the recommended improvements.

**2.1 General comments :**

The section titles should be more explicit and informative. For instance CUAS MPI / model / workflow are a bit short as section titles. The english language needs also improvements.

We changed the titles to provide more information about the relevant sections and reviewed the text.

**2.2 Specific comments:**

**2.2.1 Introduction**

– l 23 – 24 – 25 : " For simulating large areas, like entire ice sheets in adequate spatial resolution and in a temporal resolution to allow to represent changes on short temporal change, such as seasonal melt water input, an efficient numerical code is indispensable. " Clumsy and unprecises ('adequat' sufers of vagueness — adequat for what ?). To be rephrased.

We will reform the introduction will be rephrased and extended to also address a number of issues raised by RC2, e.g. "state of the art" for hydrological modelling.

– l 26 – 30 : OK but it is not said properly – the number of time steps depends on the length of the time step and of the time interval to be simulated. Maybe better to talk of 'time discretization' rather than 'time step'? + english problems : an efficient code.

We adopted the referee's suggestion.

– l 33 : compute clusters → supercomputers.

We adopt the referee's suggestion.

Add parenthesis around the reference to Balay et al.

We adopt the referee's suggestion.

– l 37 : repetitions (then ... then)

We rewrote parts of this block.

**2.2.2 Section 2**

– l 41 : The title of section 2 should be changed : a simple acronym cannot be a proper section title. At least the full name of the CUAS-MPI tool should be detailed alongside with its nature (e.g. : modeling tool for sub-glacial hydrology).

We adopted the referee's suggestion.

– l 51 – 52 " Areas with high permeability represent very efficient water transport, while low permeability represents an ineffective water system. " A bit tautological. Please be more precise (e.g., areas with high permeability are associated with higher density of channels or something like that, more descriptive.)

We will reformulate this sentence to be more descriptive.

– l 55 – 56 : " To this end, an unconfined layer is incorporated, capturing the dynamics if the head is falling below the layer thickness allowing for further water drainage. " A little explaining figure (a sketch with both configuration, confined and unconfined layer) would be helpful for the reader and would enhanced the self-consistency of the paper. See also comments of the Appendix A.

The focus of this manuscript is about the performance of the new parallel MPI implementation of the often-cited CUAS physics described in (Beyer et al., 2018). We have extended Appendix A but otherwise we think it most appropriate for readers unfamiliar with the model specifics to read Beyer's paper rather than us trying to summarize relevant parts, as the issue of relevance is highly dependent on the reader's background.

– l 58 : " an evolution equation for the hydraulic head for Darcy flow " : a governing PDE for hydraulic head spatio-temporal variations, analogous to the diffusivity equation for groundwater.

We would like to keep our formulation. The term "evolution equation" is very well established and we would like to preserve the connection to Darcy flow.

– l 59 – 60 : " A second major equation is describing the change in transmissivity with time based on melting, creep and cavity formation. " This is a constitutive law for the hydraulic transmissivity of the equivalent porous medium, right ?

We have only two state variables in the model (head $h$, and transmissivity $T$). All other quantities can be derived from them using equations outlined in Beyer et al. (2018). We solve the evolution equation for $h$ using a fully implicit scheme and rely on the linear equation solvers from PETSc for this task. In contrast to this, we use an explicit time step to update $T$. The evolution of $T$ is thus not an constitutive law — it's our very simple model to account for subglacial channel evolution.

– l 62 : reference to Appendix B would be better placed in the " Software-Design " section (once again the title should be improved). Besides, given its shortness, I think it could be introduced directly in the body of the text.

Appendix B has been included in Section 2.

– l 62 : Since equations of the Appendix A are refered to explicitly in figure 1, I think that they should be introduced directly in the body of the text.

Appendix B has been included in Section 2.

– Figure 1 : How are derived the flux and effective pressure ? Operational links (e.g. : arrows) between the boxes would improve the schematization of the modeling tool I think.

This figure is intended to illustrate the physics solved in the model (left side) and how this is solved in principle using high-level pseudo-code (right side). The information about how the flux and the effective pressure is computed (based on hydraulic head, transmissivity, and geometry) is not explicitly stated in Beyer et al. (2018), unfortunately. Nevertheless, we think this information is too detailed to fit the purpose of the figure. We would prefer to keep the figure as is, with slight modifications in the terminology. We will also include the missing equations in the model description.

– l 66 : " such as Greenland " : Not specific enough. You mean probably " such as a set up for modelling the sub-glacial flows under the whole Greenland Ice Sheet " ?

The sentence has been rephrased accordingly.

– l 70 – 71 : " Time stepping parameters are optionally described by command line parameters or a time step file. " What time stepping parameters ? Is this only time step length, or is there any adaptive time step strategy ? In the first case, I don't feel that this sentence is really necessary.

We clarified the description.

– l 72 : The MPI version uses " the same command line parameters " than the serial one. If you specify it, then you should add here a reference for finding the serial command line parameters.

We specified how the list of parameters can be obtained.

– l 74 : " We use the well-known PETSc parallel math library " Here a reference for that tool should be given.

We will add an additional reference to PETSc.

– l 82 : "if the problem size allows for comparison." Why do you say that? Please explicit what you have in mind.

Reprased the sentence to clarify.

– l 86 – 87 : "which provides an uniform interface of the features we require in CUAS-MPI." Unclear, to be rephrased.

We adopt the referee's suggestion.

– l 92 – section 2.3 Workflow : this section could be put in Appendix I think.

We adopt the referee's suggestion.

**2.2.3 Section 3**

– l 136 – 137 : I don't understand why the symbol infinity is used here. The boundary conditions are defined at the boundaries, and if using the method of the image wells these bopundaries should be at finite distances of the real well ?

Unfortunately, this notation raised confusion. It was meant to say that we have Dirichlet conditions at the southern and western boundaries, while Neumann conditions are applied at the northern and eastern boundaries. We will state this explicitly in the text and provide a figure for the setup.

– l 141 : equation 1 : Always give the units of all used variables and parameters.

We will change the text accordingly

– l 121 section 3 : this section should be rewritten by splitting it in two parts, one for the confined case, and one for the unconfined ones. A figure presenting the considered geometry (position of boundaries and of image wells) should be added.

We provide an additional figure showing the setup. Nevertheless we keep the section about constructing the solutions —

[Figure]

**Figure 1.** Pumping test configuration, with Dirichlet conditions at the southern and western boundaries, and Neumann conditions at the northern and eastern boundaries. The image well configuration outside of the domain is used for the analytical solutions, which can be extrapolated to achieve the F1962 (Ferris et al., 1962) bounded-low, -med, -high solutions, respectively.

using image wells — rather short, because this is beyond the topic of this manuscript. We already cite Ferris et al. (1962) for the implementation details.

**2.2.4 Section 4**

- l 164 – 165 : " In case of a land terminating margin, the boundary condition is no flow ". Then rivers coming out of subglacial springs are neglected, right ? Adding a sentence about the consequences of this assumption would be interesting.

  This is right. Outflow boundary conditions (non-homogeneous Neumann boundary conditions) for rivers coming out of the subglacial environment are neither considered in the setup nor in the model. Discharge through those rivers would lower the hydraulic head (water pressure) in the subglacial system. Outflow could be indirectly simulated by applying Dirichtlet conditions for the hydraulic head at those river locations and some other locations, e.g. ice-marginal lakes. The model would be capable of doing so, but to our knowledge, no data set of the river and ice-marginal lake locations for the Greenland Ice Sheet exists, unfortunately. Even though outflow along the land terminating margin is not considered, and this a very strong limitation to the hydrological modelling, we think the setup is realistic enough to draw conclusions about the model performance.

  We will include this information in the manuscript.

**2.2.5 Section 5**

- l 193 – 194 : " high-throughput simulations ". I don't know the throughput concept. It might be usefull to remind it.

  In the current manuscript version it is explained in the throughput section. We will adapt to explain it earlier.

– l 198 – 199 : " We employ GCC 11.2, Open MPI 4.1.4, PETSc 3.17.4 and NetCDF 4.7.4 with HDF5 1.8.22 for our performance experiments. " Please give the nature all these software (e.g. : the compiler GCC, the message-passing library OpenMPI 4.1.4, etc)

We added information about all packages we used.

– l 202 : please give the topology of the network (e.g. : hypercube, fat tree, etc).

We added information about the fat tree topology.

– Figure 3 : please add a scale for the shaded bed topography.

This figure was meant to mainly show the mask. The shading was applied only to give the figure a more three-dimensional appearance. As this information is not relevant for the manuscript, we provide an updated version of the figure without shading.

– l 213 : " full, half, quarter and one-eighth occupied compute nodes " I think that using populated instead of occupied would be more idiomatic → full, half, quarter and one-eighth populated computing nodes.

We adopt the referee's suggestion.

– l 216-218 : " The size of the circle indicates the hardware investment. The smallest circle indicates that only one node was used, the next size up indicates two nodes, then four nodes, and lastly 16 nodes. " This should also be specified directly on Figure 4.

This will help understanding the figure and we will add the explanation to the capture.

– l 219 – 224 : I don't understand the runtimes mentionned in the text. For instance 4900 s is stated for 12 threads on 1 node, while on Figure 4 the vertical coordinate of the corresponding point is 140 s. 4900 is not a CPU time, since 140 s *12 cores = 1680 cpu.s (or 140*96 = 13440, if we consider that the whole node is requested for the computation even if only 12 cores are really used), which is different from 4900.

Currently the numbers does not match. We will fix this and apologize for this unawareness.

– l 225 – 232 : Very interesting discussion, which concludes to the interest of half-populated runs with CUAS-MPI on the considered cluster. I think that the CPU times associated to the slurm requests (→ twice bigger for a half-populated run compared to the fully populated one for a given number of MPI processes - if the run times are equal !) should be also discussed.

We will extend our discussion on this topic.

– l 234 : I think that 'parts' would be a better word choice than 'categories'.

We adopt the referee's suggestion.

– l 249 : " which writes a single output of configuration "large" " : please say here what is the total size (in bytes) of the outputs written in each case by NetCDF.

We will at this, as it will help to get the context correctly.

– l 254 : " scales up to 2304 MPI processes " : according to Figure 5 I would rather say that it scales up quasi-linearly ut to 1536 MPI processes.

We adapted the text to this comment.

– l 259 – 262 : " In particular, we see approximately linear scaling also for the "CUAS-MPI system kernels" and the system matrix routine up to at least 768 MPI processes. Then the "CUAS-MPI system kernels" and thereafter the system matrix creation reach their scaling limit and their respective runtime increases." This is correct for CUAS-MPI system kernels, but not for the system matrix creation, which well scales up to 3072 MPI processes.

We adapted to a more precise description.

– l 264 : " grid data exchange communication caused by PETSc " : I think that this term is misleading, and so is the associated legend in Figure 6. The problem as far as I understood is not the kernel computations, but the communication to PETSc of the results obtained by these kernel computations, so that the PETSc linear solver can use them for solving the diffusivity equation governing the sub-glacial flow. So I would keep 'kernel computations' for the green curve in Figure 6, but use 'kernel communications to PETSc' instead of 'PETSc communication' for the pink one.

The CUAS kernel operates on data structures from PETSc. So in performing the kernel computation, functions from the PETSc library are called, for example, to handle border exchanges. We follow the reviewer's suggestion and use the term "kernel communication" in the legends, and will clarify this issue in the text.

– l 267 – 268 : " In our studies of throughput, i.e. how many simulated system years we can run in a day of compute time (simulated years per day, SYPD). " I didn't know this concept of throughput as a measure of result production rate – which does not mean that it does not already exist in the literature! But still I think that because it has been already used earlier in the manuscript (even in the abstract), its definition should have been explicited (or reminded) earlier as well.

We will place this earlier.

– l 278 : The observation of a significant impact of I/O rate on computation time is interesting. This question has for instance also been investigated in another porous media related context in Orgogozo et al., 2014 (http://dx.doi.org/10. 1016/j.cpc.2014.08.004 see section 4.4).

The oberservation of Orgogozo et al. is similiar to our findings. We will consider it.

– Figure 7 : the letters a, b and c used in the caption are missing on the figure.

We will add letters to other Figures to and emphasize it in this figure.

– l 284 – 288 : This paragraph is hard to follow. I don't think that one can say that larger grids need more cells per MPI process than smaller grids for efficient parallelism. While using larger grid, one increases the computational load and thus the computation over communication ratio, which leads to better parallel efficiency for a given number of MPI processes.

Our statement is related to the fact, that the graph uses "cells per MPI process" on the x-axis. We will clarify this.

190 And also I don't understand what is the sweet spot.

In this context, we assume that the sweet spot is the minimum runtime and will clarify this in the text.

– l 291 – 292 : "The minimum of the total runtime per iteration is increasing with spatial resolution". Please discuss why?

We will explain it. It is related to the metric "cells per MPI process".

– l 292-293 : " the spread is less than the total runtime per year, showing that the total runtime is driven by the increase in
195 number of linear solver iterations with resolution. " : Hard to follow. Does panel b of Figure 7 present the total runtime
per year ? If yes say it explicitly in the caption of the figure and refer explicitly to this figure in that discussion.

We rephrased the description in order to clarify the issue.

**2.2.6 Section 6**

– l 295 : "but potentially not highest resolution." Unclear.
200 Currently the highest resolution is 150m as this is the current bedrock resolution. We will add this information in the
text.

– l 297 : " the amount of core-hours usually available for such runs. " Be more precise. What is the target of CPU hours
for this kind of run, and why ?

This sentence is unfortunately not precise, and we would drop the second part of the sentence: "Although ... limits
205 the number of years that can be simulated." The number of core-hours strongly depends on the scientific question. If
someone wants to run the hydrology model as part of an Earth-System-Model over decadal to millennial time scales, the
computational resources spent just for the subglacial hydrology would probably be less, than for a detailed simulation
with very high resolution (time and space) for e.g., hydropower plants.

– l 297-298 : " A simulation covering the 90 years from 2010 to 2100 in 600 m spatial and 1 hour temporal resolution
210 requires a wall-clock time of 1350 hours (56 days) on 384 MPI processes. " Please precise again here that this quantifi-
cation is obtained for the Lichtenberg HPC system.

We clarified it at this point.

One could discuss how it could be extrapolated to larger supercomputers, for instance European level ('tier-0') ones.

The data show that already on the Lichtenberg system, CUAS-MPI reaches its scaling limits, so using even bigger
215 computers is not a productive use of such resources.

– Section 6 : english langage problems ('might of interest', 'now have now', ... )

We checked the entire section again.

– l 322 : G250 is too cryptic ; either add a more informative name or no name for this resolution.

We will add a brief description to G250 to give the reader context.

220  – l 326-333 : This discussion is a bit confusing, to be rewritten more clearly.

We adopt the referee's suggestion.

Add also physics-based arguments for the statement that there is no necessity of high frequent data exchange.

The question on how often to couple both models depends on the phenomena that are investigated in both models. If someone for instance wants to study the effect of ocean tides on the subglacial environment, time steps in the hydrology

225  model needs to resolve the tides. If the ice sheet model can afford the same time steps in terms of computational costs, why not? But in most cases, the ice sheet model will be computationally more expensive and a compromise must be found (e.g., daily coupling only). We will rephrase this sentence.

– l 345 : " computational granularity " : please remind briefly the definition of this concept here

We will add an brief explanation: granularity is a measure of the amount of work which is performed by a task.

230  ## 2.3  Appendix A :

We will incorporate Appendix A into the main text as suggested by both reviewers. This part will also be extended slightly to address a number of issues listed below. We expect that this will also improve the clarity in the validation section.

– Psi and b should be introduced in a little explaining figure as proposed above (see also the comment of lines 55–56).

Both reviewers ask for more information about the meaning of quantities in the hydrological system. We will briefly

235  state, what every quantity is referring to.

– The units of all the used variables and parameters should be given.

Yes, we fully agree.

– l 376 : " To allow a smooth transition between the confined and unconfined system, a range d is introduced. " This parameter should be a bit more discussed. What is its physical signification? What is its range of possible values?

240  The parameter $d$, with $0 \leq d \leq \Psi$ (unit: m), allows for a gradual transition of the effective storativity, $S_\mathrm{e}$ in the confined-unconfined-transition scheme (Eq. A2 in the manuscript and Ehlig and Halepaska (1976, Eqns. 6, 7 and Fig. 2). We will move this sentence and the additional information below Eq. A2. In all the simulations used here we use $d = 0$ as in (Beyer et al., 2018, Tab. 3).

– l 382 : The difference between Ss and Sy should be more explicited.

245  We will give additional information about all hydrological quantities including $S_\mathrm{s}$ and $S_\mathrm{y}$.

– l 387 : There is surely (at least) one publication associated with this complex equation ; please refer to it here.

We cite Ehlig and Halepaska (1976) as this is the foundation for the confined-unconfined scheme used in Beyer et al. (2018) and implemented in CUAS-MPI.

– l 392 : there should be a link between h and pw ? How pi is computed, or from where does it come from if it is a forcing

250  ? Please explicit these point here.

The hydraulic head is linked to the water pressure via $h = p_w/(\rho_w g) + b$, with the density of water $\rho_w = 1000 \, \mathrm{kg \, m^{-3}}$), the acceleration due to gravity $g = 9.81 \, \mathrm{m \, s^{-2}}$, and bed elevation $b$ (unit: m). The ice overburden pressure $p_i = \rho_w g H$, with the bulk density of ice $\rho_i = 910 \, \mathrm{kg \, m^{-3}}$ and the ice thickness $H$ (unit: m). The ice thickness is taken from the BedMachine Greenland dataset (Morlighem, 2021; Morlighem et al., 2017, Version 4) as part of the model model input.

255 ## 2.4 Appendix B :

– l 399 – 400 : 'equidistant' ; may be regular would be more appropriate ? And why using rectangular grid cells instead of square ones ?

We adopt the referee's suggestion.

– l 405 – 406 : " If an iterative solver is used, convergence is decided by the decrease of the residual norm relative to
260 the norm of the right hand side (rtol) and the absolute size of the residual norm (atol). " Here an equation allowing to precisely identify the quantity rtol and atol should be provided. Typical ranges to be used should also be given.

We don't think, that typical values for convergence values exist. They usually dependent on the physical problem to be solved, the boundary conditions and also the initial conditions in a specific setup. We state the values that we have used already in the main text and we will provide brief information about the tolerances.

**References**

Beyer, S., Kleiner, T., Aizinger, V., Rückamp, M., and Humbert, A.: A confined–unconfined aquifer model for subglacial hydrology and its application to the Northeast Greenland Ice Stream, The Cryosphere, 12, 3931–3947, https://doi.org/10.5194/tc-12-3931-2018, 2018.

Ehlig, C. and Halepaska, J. C.: A numerical study of confined-unconfined aquifers including effects of delayed yield and leakage, Water Resources Research, 12, 1175–1183, https://doi.org/10.1029/WR012i006p01175, 1976.

Ferris, J. G., Knowles, D. B., Brown, R. H., and Stallman, R. W.: Theory of Aquifer Tests, Tech. rep., U.S. Government Print. Office, https://doi.org/10.3133/wsp1536E, 1962.

Morlighem, M., Williams, C. N., Rignot, E., An, L., Arndt, J. E., Bamber, J. L., Catania, G., Chauché, N., Dowdeswell, J. A., Dorschel, B., Fenty, I., Hogan, K., Howat, I., Hubbard, A., Jakobsson, M., Jordan, T. M., Kjeldsen, K. K., Millan, R., Mayer, L., Mouginot, J., Noël, B. P. Y., O'Cofaigh, C., Palmer, S., Rysgaard, S., Seroussi, H., Siegert, M. J., Slabon, P., Straneo, F., van den Broeke, M. R., Weinrebe, W., Wood, M., and Zinglersen, K. B.: BedMachine v3: Complete Bed Topography and Ocean Bathymetry Mapping of Greenland From Multibeam Echo Sounding Combined With Mass Conservation, Geophysical Research Letters, 44, https://doi.org/10.1002/2017gl074954, 2017.

Morlighem, M. e. a.: IceBridge BedMachine Greenland, Version 4, https://doi.org/10.5067/VLJ5YXKCNGXO, 2021.

---

## Author Comment (AC2)

**GMD Reviews and Authors' Response concerning the paper "A parallel implementation of the confined-unconfined aquifer system model for subglacial hydrology: design, verification, and performance analysis (CUAS-MPI v0.1.0)"**

Yannic Fischler[1], Thomas Kleiner[2], Christian Bischof[1], Jeremie Schmiedel[4], Roiy Sayag[4], Raban Emunds[1,2], Lennart Frederik Oestreich[1,2], and Angelika Humbert[2,3]

[1]Department of Computer Science, Technical University Darmstadt, Darmstadt, Hesse, Germany
[2]Alfred-Wegener-Institut, Helmholtz-Zentrum für Polar- und Meeresforschung, Bremerhaven, Bremen, Germany
[3]Faculty of Geosciences, University of Bremen, Bremen, Germany
[4]Department of Environmental Physics, BIDR, Ben-Gurion University of the Negev, Sde Boker, Israel

**Correspondence:** Yannic Fischler (yannic.fischler@tu-darmstadt.de)

*Copyright statement.* ©2023 all rights reserved

**1 General Comments**

We are grateful for all suggestions concerning the structure of the manuscript that will help to improve the manuscript - we will incorporate the vast majority of the suggestions, including to move sections to the appendix. We are also happy to include more text in the introduction. We will extend the entire description of the state of the art of hydrology, the description of the hydrology of this model and the description of the modeling of the pumping tests.

We want to emphasize that CUAS-MPI is a code that has been developed from scratch. Its software architecture and numerics are different from former implementations of the same equations of the hydromechanical problem. The acronym CUAS stands for Confined-Unconfined Aquifer System, which both, the prototype code CUAS and CUAS-MPI are simulating. It is comparable to groundwater equations being solved by numerous different codes, ranging from very basic serial implementations to performance optimised codes for HPC.

We also want to lay out, that the model (and code) that the paper Fischler et al. (2022) was concerned with, is an entirely different model simulating another compartment of the Earth system, namely ice sheets. Consequently, none of the conclusions drawn in our manuscript can be drawn from Beyer et al. (2018) nor from Fischler et al. (2022).

**2 Review 2 by an anonymous referee**

**2.1 General comments:**

The manuscript by Fischler et al. discusses the development and performance of an MPI-aided parallel implementation of a pre-existing subglacial hydrology model (Beyer et al, 2018). The method and some specificities of the parallel implementation are briefly presented (including its reliance on external libraries such as PETSc). The model is then validated using several test cases, some of which possess an analytical solution. Finally, runtime results for a series of bigger test cases (the size of the entire Greenland ice-sheet) are analyzed in order to derive tendencies concerning the scalability and throughput of the modeling tool, with the intent of comparing these to the performance of the well-known ice-sheet model ISSM (in view of a future coupling).

Overall I would not recommend this paper for publication in its present format. The development of an efficient subglacial hydrology modeling tool capable of scaling on supercomputers is of interest to the community; so there is no doubt that CUAS-MPI has scientific relevance. However, the overall quality of the manuscript is poor and the scientific significance and originality of the results presented is only marginal. To be considered for publication, improvements should concern how the paper is structured –to be able to better distinguish what will be discussed in each Section; but there also should be a rethink of the general content. Indeed, my biggest concern is that the manuscript does not present any novel concepts or introduce any new methodologies and that most conclusions drawn can already be found in previous work (see Beyer et al. 2018 and Fischler et al. 2022). The English quality also makes the manuscript difficult to read at times and many sentences/explanations would benefit from a rewrite. Finally I do not believe that the Introduction is complete with all relevant citations and references.

Due to the large number of major concerns, I have gathered all specific comments and technical corrections below, following the current structure of the paper. For each Section, I start with a paragraph summarizing my concerns before I provide more technical, line by line, suggestions.

**2.2 Specific comments:**

On a general note and as suggested by reviewer #1, I would strongly advise making the titles of Sections more informative. We will adapt the titles.

**2.2.1 Abstract**

– The abstract suffers from a lot of repetitions (CUAS-MPI).

– Acronyms (such as MPI) should be detailed at least once.
   We will elaborate acronyms such as GCC, MPI and PETSc

– Beyer et al. 2018 should be cited.
   We are citing Beyer et al. (2018) intensively in the entire manuscript, but we prefer to keep the abstract free from citations.

- l. 1 – 2: what is meant by "as well as marginal lakes and rivers" ? Rephrase this sentence

  "The subglacial hydrological system affects . . . ice marginal lakes and rivers". The subglacial water can feed into ice marginal lakes and rivers. Not only into the ocean. We will rephrase acordingly.

- l. 2: "has been developed" makes it sound like CUAS is new, when this tool is pre-existing. Rephrase.

  As outlined above, the CUAS acronym stands for "Confined-Unconfined Aquifer System" in general and the physics have been implemented in a new code called CUAS-MPI. We will rephrase this sentence here and elsewhere in the manuscript to make the distinction between "CUAS" and "CUAS-MPI" more obvious.

- l. 7: repetition of throughput could be avoided

  This is correct. We rephrased the sentence to "Our measurements show that CUAS-MPI reaches a throughput comparable to that of ice sheet simulations, e.g. the Ice-sheet and Sea-level System Model (ISSM)."

**2.2.2 Introduction**

My main issue with the introduction is that it lacks a proper state of the art. Context about the need for subglacial hydrology modeling should be provided, and different existing modeling approaches/proper citations for other similar modeling tools should also be acknowledged.

The place and goal of the present study are also not clearly stated and should be elaborated upon. Finally, a sentence underlining the main findings should be included.

We will elaborate upon the scientific goal(s) in the introduction, but prefer to state the main findings in the Abstract and Conclusion sections only.

- l. 13: consider giving a ref to accompany the first sentence (especially for Antarctica)

  This is a good point indeed. We have now gone this route: we provide a number of papers on observations of the impact, although this is not trivial as not many of such observations do exist. We cite now one for ice stream motion (Engelhardt and Kamb, 1997), one for tidal modulation (Bindschadler et al., 2003), one on seasonality of ice sheet motion due to meltwater input (Hoffman et al., 2011) and one on subglacial lakes (a review by Siegert (2000)). There is a by far larger number of papers on simulating ice stream flow and the effect different type of sliding has on the ice sheet motion, but we think it is useful in this very first sentence to stick with observations.

- l. 15: use the same units (choose ma-1 or mmd-1)

  We adopt the referee's suggestion.

- l. 19: "hundreds to thousands of metres thick ice" → "meters of thick ice"

  We adopt the referee's suggestion.

- The sentence starting at the end of l. 20 should be rephrased, and the citations should be under (). I suggest transforming this sentence into a full paragraph with a proper state of the art of subglacial hydrology modeling.

Although the main topic of the manuscript is the performance of the new implementation CUAS-MPI of the published methods (Beyer et al., 2018), we are happy to add a paragraph about the state of the art of subglacial hydrology modeling, as suggested.

- The sentence starting at the end of l. 23 is too vague, consider quantifying "adequate spatial resolution" and "temporal resolution"

  We omitted this sentence as we think the next, rephrased, paragraph motivates our work.

- l. 26: same issue, consider quantifying "the desired amount of timesteps"

  We rephrased the paragraph to clarify the motivation for this work.

- l. 27: " An example of relevance is the projection ... until 2100..." → why is that? (consider providing a reference)

  Ice sheet dynamics is contributing significantly to sea level rise and projections show that this is going to increase in the next centuries. These projections are typically done up to 2100 as climate forcing for such simulations is available for this time period. As the subglacial hydrological system is having a big influence on the sliding of ice over the bed, projections of the contribution of ice sheets to sea level change will benefit if for the same time period simulations of the subglacial system are available. This is why we refer to this time period. However, indeed all ice sheet simulations will benefit from simulations of the subglacial system, paleo-simulations over millions of years similar to high resolution simulations of ice caps close to settlements. We are happy to provide a reference and extend the sentence in the manuscript to make this point more clear.

- l. 30: "This can only be achieved with a efficient codes"

  We rephrased this section of the text.

- l. 31: again and unless I am missing something, CUAS is not new. The use of "developed" is misleading here.

  We want to emphasize that CUAS-MPI is indeed an entirely new developed code. While the set of equations to be solved is similar to CUAS, the numerics is new, the software architecture, the parallelisation - as such the entire software is coded from scratch. Therefore, the use of 'developed' is indeed correct here.

  I suggest merging this short sentence with the following one.

  We adopt the referee's suggestion.

- l. 33 "compute clusters" → supercomputers?

  We adopt the referee's suggestion.

- l. 34-35: provide more information about these "pumping tests". A short description with a proper reference would do.

  Indeed this is a very good suggestion. We will include a paragraph on the pumping test in the introduction with proper references.

- l. 35: "we employed CUAS-MPI then" → "we then employed CUAS-MPI"

  We adopt the referee's suggestion.

    – l. 36: perhaps expand on the "performance data" that you gather and provide a short description of the nature of Score-P

We expanded a half sentence to clarify the type of data collected and will add a paragraph on Score-P in the experimental section.

    – l. 37: "first we explain the underlying model ..." → "We explain the underlying model in Sect. 2"?

**2.2.3 Section 2**

115 Section 2 appears a bit disorganized and could benefit from a complete makeover. The purpose of each subsection is not clear to me. Based on what I understand, I believe Appendix A should replace most of Section 2.1 (see below).

Thanks for the suggestion. We will move and extend Appendix A into Section 2. We expect this will improve the connection to the Validation section(pumping tests) and avoids duplications of text introducing certain quantities, symbols and units.

It also sounds like Section 2.2 could be a good place to describe your original work: how did you make the pre-existing
120 CUAS solver better, what did you change/improve?

Again, we want to emphasise, that CUAS-MPI is a new code, whereas CUAS was only a prototype or proof-of-concept code. But we do understand the intention of the reviewer and we will add more text to make clear what advantage a professional numerical code has. However, the description provided is too vague to be impactful (e. g. l.74-90: "the distributed memory features of PETSc", "a PETSc feature", "list of available PETSc solvers", "the iterative GMRES solver", "an adapter which
125 provides a uniform interface to the features we require" etc.). I suggest rewriting this entire section with appropriate references to the external libraries (PETSc, NetCDF, HDF5) that you use, including a more specific description of the features that you select and why you chose to do so. Consider providing a more detailed layout of the code and its submodules in Fig 1 to accompany these developments

We have added a paragraph that outlines our design philosophy and the resulting choice of PETSc and HDF5: *The goal of*
130 *our code development was to arrive at a software artefact that is performant on current HPC systems but also maintainable in the light of future HPC system evolution. To that end, we based our development on the well-known PETSc parallel math library. PETSc is an open source software supported by a wide user base and, for example, part of the software stack supported by the U.S. exascale project https://www.exascaleproject.org/research-project/petsc-tao/. Similarly, to parallelize the NetCDF output, we used the HDF5 library, an open-source product maintained by the not-for-profit HDF group. By basing our*
135 *development work on these software infrastructures, we profit both from their maturity and performance on current systems, but also, in particular, from performance improvements that will be implemented by the community supporting these software libraries down the road.*

The purpose of Figure 1 is to show what physics is being solved (left side), and a high-level pseudo-code (right side) of how this is being done. The overall numerical approach is unchanged to that of Beyer (Beyer et al., 2018), except for the use of an
140 iterative solver for the linear systems of equation. As our paper focuses on the parallel implementation and its capabilities, we do not consider it appropriate to duplicate this information here. However, we will revise this section to provide more details on the libraries and their relevant features.

Finally, the point of Section 2.3 is also a little bit difficult for me to understand. I do not follow what is the relation between, and purpose of, the various scripts mentioned. Here also, I cannot differentiate what is original work when compared to the pre-existing CUAS workflow. I suggest rewriting this subsection, and accompanying the text of l. 109 - 118 (where "typical use cases" are employed as examples), perhaps sketching the workflow?

The workflow for the python implementation of CUAS and the new implementation, CUAS-MPI, is very similar, because it is mainly pre-processing available datasets and setting up the experiments. This part of the manuscript is intended as a suggestion and illustration of a workflow that worked for us and that could be starting point for new users. Nevertheless, this part seems to distract from the main line of thoughts and we move this to the Appendix section.

**Section 2.1:**

- l. 43: "different perspectives on the purpose of" → "different reasons for"? This sentence is awkward and could use a couple references to illustrate your point.

  Changed to "different reasons for" as suggested. The initial (mostly descriptive) part of section 2.1 will be merged into a 'state of the art' paragraph in the introduction.

- End of l. 45, starting with "In the past . . . " is odd in this context. If a proper state of the art is placed in the Introduction, it should be removed altogether.

  We agree and make changes accordingly.

- l. 47: "the code that we present" → "the code that we employ/use"

  We do indeed present a new code here, this is why we do not agree to switch to employ/use.

- l. 50: "the void space is fully saturated" is confusing when it is said later in l. 55 that "it may happen that water supply is not sufficient to keep the water system fully saturated". Please clarify.

  This is exactly what the confined-unconfined scheme is about. If not sufficient water is available to have a fully saturated system over the entire thickness of the layer, the saturation thickness of the layer ($\Psi$) is reduced, while the system below that thickness remains fully saturated - this is the unconfined aquifer. To be more explicit: The saturation thickness is the depth from the water table to the base of the aquifer.

- l. 54: "inefficient water transport", but the entire sentence should be rephrased to be more impactful and precise.

  Very good suggestion. We rephrased this to: "The ice sheet is acting as a confining layer of the aquifer. It may happen that water supply is not sufficient to keep the water system fully saturated. To this end, an unconfined layer is incorporated, capturing the dynamics if the head is falling below the layer thickness allowing for further water drainage."

- l. 58 - 63: As suggested by reviewer 1 I would strongly suggest moving Appendix A here, along with providing proper citations (to Beyer et al 2018 and de Fleurian et al 2014). Describing equations is always a difficult task, and given the length of Appendix A and the fact that the equations are referenced in the body of the text I do not believe it would make the paper unnecessarily longer (especially if the rest of Section 2.1 is rewritten in a more concise manner). Consider

175  using an image to describe the model? Or at least a reference to Fig. 1 of Beyer et al 2018.

  We agree.

**Section 2.2:**

 – l. 66: "its performance was too low for larger setups such as Greenland" is too vague. Please clarify

  We will explain this from the viewpoint of a domain scientist.

180 – l. 68: please provide a description of the "physics kernels"

  Many thanks for this feedback. We originally thought that Figure 1 is giving this overview, but we learned that this is not detailed enough. In the revised version we will add a paragraph to elaborate more on the physics kernel.

 – l. 74: provide a reference to PETSc

  We will add an additional reference to PETSc - please note the PETSc has been cited already.

185 – I feel like Table 1 and the associated discussion in Section 2.2 (l. 87-90) is unnecessary

  In the usage scenarios we encountered so far, the output and output options were relevant.

**Section 2.3:**

We moved this section to appendix.

 – l. 96 – 97: "a setup script', "the mask", "any environment"... please be more precise

190  We are not sure we understand what exactly the reviewer means. In the sentences before we detail what the setup script is actually for: 'A typical workflow to run a simulation is described by a setup script. In that script the model domain and the grid are defined 95 and used to create the mask including boundary conditions. This mask is one of the input fields.' And then we explain in which environment you can perform this - on any environment, a local cluster, a desktop computer, a laptop or a HPC cluster. We are happy to improve our text once we understand what exactly the reviewer is

195  referring to.

 – Usefulness of l.100 – 104 is questionable (restarting a run is common practice)

  Indeed, restarting a model run is common practice. Here we try to explain, that the restart capability of CUAS-MPI, and also the former python implementation, can be used in alternative ways. E.g., restarting from a coarse-resolution spinup after remapping onto a high-resolution mesh.

200 – l. 103: "has be remapped" → "has been remapped"

  We adopt the referee's suggestion.

 – l. 111 – 112: "Here the seasonal . . . and others with seasonal forcing" should be rephrased

  Weill be rephrased. "At this point the actual simulation with seasonal forcing starts using the refined mesh. It is highly recommended to also run a control simulation on the refined mesh by further using the stead-state forcing. This allows

205  to account for model transients that may still be contained in the model after the spin-up."

– l. 113: can you elaborate on this "nesting approach" mentioned?

As this part of the manuscript (workflow) is of only minor relevance for the main topic (performance) we will only briefly elaborate on this topic. A regional subdomain nesting for a particular area of interest can be used to reduce the computational costs of a model run. If a drainage basin can only covered partially it is very beneficial to embed this area into a large-scale model run that provides boundary conditions along the subdomain margin.

– l. 120: what is the meaning of that last sentence? (what are "these numbers"?)

After restructuring Section 2 and moving the 'workflow' section into the appendix, the last two sentences of this section are no longer required.

**2.2.4 Section 3**

This Section is interesting and the work conducted is solid but my impression is that the CUAS-MPI equations and numerical methods employed to solve them have not changed since what is described in the paper by Beyer et al., 2018. This makes the relevance of the proposed validations questionable.

Indeed, the equations have not changed in the CUAS-MPI implementation since Beyer et al. (2018), but the equations have also not been tested with pumping tests before. Nevertheless, there is a need to show that the new implementation leads to expected results. We think, that pumping tests are very good test cases, because they are very well established in the field of hydrology. Specifically, analytical and semi-analytical solutions exist, which CUAS-MPI does compute correctly. With the pumping tests, one can verify the implementation of the governing equation (Eq. A1), as well as the implementation of Neumann and Dirichlet boundary conditions. They are further relatively easy to set-up and conduct.

I would suggest either to elaborate on the relevance of these test cases, describing them in more depth, or moving them to an Appendix.

We probably cut this part of the manuscript too short in the submitted version. We think pumping tests are very important and extend this part of the manuscript.

If kept, consider providing a Table summarizing the various test cases performed (from the body of the text it is unclear how many tests are actually designed), and clearly identifying what code feature they are testing.

The experiments conducted are shown in Fig. 2 and we don't think a table is needed with the extended version of this section.

Finally, I cannot emphasize enough how much a sketch or a picture would help the reader understand what these "pumping tests" are.

This was also recommended by RC1 and we are happy to provide a new figure illustrating the set-up including boundary conditions, distances and image-well types and locations.

There is a lot of math involved here that is not trivial, and a proper introduction of the terminology would be helpful.

We moved the equations from Appendix A to the main text in section 2, as suggested by RC1 and RC2. Now, most of the quantities and units are already mentioned prior to this section.

– l. 127: consider providing a more detailed description of these "pumping tests".

We agree and will provide a more detailed description in the revised version.

– l. 134: I am unfamiliar with the method of images in this context, I believe a short justification or a reference would be helpful

The justification is that the partial differential equation for the confined case is linear, thus allowing the use of a superposition (known as the method of images), the reference is Ferris et al. (1962), mentioned in l. 135.

– l. 138: please provide a brief description of what is an "image wells"

An image well (or an imaginary well) represents the numerical boundary in the mathematical termenology. Thus, the two boundary conditions can be described with the method of images using wells projected perpendicular and equidistant to the boundaries as the "real" pump (Ferris et al., 1962). We hope to clarify this with our new figure illustrating the set-up including image-well types and locations.

– l. 140: define "drawdown"

The drawdown is already defined in l. 140 with $s = h(x,y,0) - h(x,y,t)$. Alternatively: drawdown, the amount the hydraulic head has changed since its initial state $s = h(x,y,0) - h(x,y,t)$.

– l. 145: what is "the non-linear unconfined case" ? (again a Table summarizing the test cases and giving them names would help)

We appreciare the reviewer suggestion. The simulations conducted are shown in Fig. 2 and we think with the extended version of this section we don't need a table. We will mention the abbreviations of Fig. 2 in the text directly.

– l. 153: what is "the specific yield" (another case for describing the model's equation in a previous section)?

The specific yield is the ratio of water volume per unit volume which gets released once the aquifer drains. This is only relevant for the unconfined case due to a partially filled aquifer $\Psi < b$ (Eq. A2). We moved the equations from Appendix A to the main text in section 2, as suggested by RC1 and RC2 and changed the text accordingly.

**2.2.5 Section 4**

I feel like this section should be a subsection of Section 5.

We changed the text accordingly.

It could be useful to see the various grid resolutions, perhaps as a Supplementary Material?

– l. 162: the first sentence should be rephrased.

Rephrased and split into two sentences.

– Fig 3: the bed topography is not visible, consider changing the color scheme or the transparency. Also, there should be a scale for it.

The shading was for three-dimensional appearance of the figure only. There is no scientific benefit on showing the shading with a proper scale so we update the figure without shading.

270    – l. 165: please elaborate on the "Dirichlet boundary condition for the head". A pressure is not homogeneous to the hydraulic head.

This reviewer is correct. The hydraulic head is zero at the ocean boundary and is imposed as a Dirichlet condition. This is equivalent to assuming, that the water pressure in the subglacial aquifer is equivalent to the pressure in the ocean at the same depth. We ignore the slight density difference between sea water (about $1028\,\mathrm{kg\,m^{-3}}$) and fresh water

275    ($1000\,\mathrm{kg\,m^{-3}}$). We correct this in the text.

– l. 168: the citation Christmann et al. 2021 is odd. What are you trying to back up with it?

We are referring to a version of BedMachine that has been created with new airborne data from AWI for the publication Christmann et al. (2021). We have used this new BedMachine dataset for our simulations here and there is no other way to reference it, because it is not one of the major BedMachine releases.

280    – l. 168: "To summarise it", what is "it" ? Also, I am not sure what you mean by this sentence.

The margins along the ice sheet consist of marine terminating ice masses and land terminating ice masses. The marine terminating ice masses have a different boundary condition (head boundary condition) then the land terminating ones (no flux). This way the water moves from the inland into the fjords (marine terminating glaciers) only.

– l. 171: "Ice sheet basal melt" can you clarify what you mean here, where are you using this information into your

285    equation system (what are the units) ? This sentence seems redundant with l. 182

Ice sheet basal melt (unit: $\mathrm{m\,a^{-1}}$ water equivalent)) is the freshwater source $Q$ for the model (Eq. A1). This will be clear in the revised version with Appendix A moved into section 2. We drop the last part of the sentence in l. 182, because of this redundancy.

– l. 176: remove the comma

290    We adopt the referee's suggestion.

– l. 184: you defined a symbol for the specific yield earlier, use it

We adopt the referee's suggestion.

– l. 191: remove the "residuum"

We adopt the referee's suggestion.

295 ## 2.2.6    Section 5

Section 5.1 is actually well written, and Fig. 4 is a good illustration, but the conclusions lack generality: do they hold for other node architectures? Can we identify directions for potential code improvement? It would help to more thoroughly quantify the memory requirements and their nature to answer these questions. It sounds like the point that is made is simply to provide users

with recommendations when running this specific code on this specific supercomputer, which is not extremely useful.

300  The Lichtenberg Architecture is actually very common as far as HPC systems are concerned: Intel- or Intel-compatible chips connected with a high-speed network. We have made that point in the text. So our findings are highly relevant for HPC installations other than TU Darmstadt. In the section on software design, we added a paragraph on why we chose PETSc and HDF5 as basis of our parallel implementation, as they are mature software infrastructures that are being developed further and where we will be able to profit from this work just be linking new versions of these libraries. So, in fact, our software design is

305  designed to run well across a wide variety of platforms, now and in the future.

In Section 5.2 the contributions from various pieces of the solver to the total runtime are quantified. The first comment that can be made here is that if outputting destroys the analysis that much, and you choose not to discuss this phenomenon, why include it in the curves? It is very distracting, particularly for the lower resolutions since "NetCDF output" is seen to dominate runtime.

310  Depending in the application scenario, the state of the system will be saved more or less often. So output is an important component of the overall CUAS-MPI software. Even parallel output is, however, an expensive operation, even with a state-of-the-art implementation such as HDF5, and that is particularly the case in application scenarios where there is little computation, i.e., low resolution runs. Figure 5 shows the regime where one output is written daily. In Figure 7a, performance results with daily and annual output are shown, and in the annual case the cost of output is negligeable.

315  More importantly, the conclusion here seems to be that there is a tipping point where scalability breaks, which depends on the relative costs of the various "subcategories" and the general amount of work per MPI process (G600 is experiencing scalability issues faster than G150 because it has less elements per threads). Not surprisingly, the curves presented in Fig. 5 and the explanation behind the observed tendencies are similar to what is discussed in depth in Fischler et al., 2022. The culprit for the inflexion is the same: both system matrix creation and PETSc communications experience scalability issues due

320  to the overhead costs of MPI communications (citation from the 2022 paper: "with an increasing number of MPI processes, the communication overhead of the assembly starts dominating at some point."). Unfortunately, the same can be said of Sect. 5.3: the two conclusions starting l. 279 "In general we see that for smaller grids there is no sense in using a large number of processes, as there is not enough work to be done for an efficient parallelization" and l. 285 "...more MPI processes, in general, cause more communication and synchronization overhead and more computation is needed to offset this" are similar to those

325  drawn by Fischler et al. 2022, limiting the impact and importance of this study. Is there a way to differentiate this work from the 2022 paper?

Our manuscript discusses an entirely different model than Fischler et al., 2022. Fischler et al. (2022) was about a model about ice dynamics, comprising an approximation of the Stokes equation, enthalpy balance, evolution equations for the geometry with boundary conditions. The equations are discretized using finite elements. CUAS-MPI is a hydromechanical model solving

330  basically a groundwater equation. Both models are concerned with compartments of the Earth system that interact with each other, but this is all they have in common. Both models are in the same environment, on HPC platforms, and if they are run in a coupled mode, we find it worth to show the scaling of both applications. Perhaps perspective on code improvements could be discussed?

As we present a new implementation of a model, code improvements are not the scope of this paper, but may well be a topic in the next years.

- The first paragraph starting l. 193 should be rephrased

- l. 195: "()" should be removed around Section 4

  We adopt the referee's suggestion.

- l. 201: is there a link to learn more about that particular supercomputer?

  A link to the system web site was added: https://www.hrz.tu-darmstadt.de/hlr/hochleistungsrechnen/index.en.jsp l. 203: what is "turbo-mode"?

  Explanation added. "Turbo-mode allows the CPU to exceed its base frequency for short time, but also considerably increases power consumption."

- l. 205: "relative standard deviation" of . . . ? (the time)

  Yes, indeed the time, we will be more precise at this point. Many thanks for pointing this out.

- I agree with reviewer #1 that the vertical coordinate of Fig. 4 is not consistent with the runtimes reported in the body of the text.

  We will check and fix it and apologize for this unwariness.

- l. 234 - 242: I would refrain from using too many "", as they clutter the text.

  We agree and will use italics instead.

- The terminology is awkward at times: these so-called functional categories should be put in the context of a workflow akin to Fig. 1 in Fischler et al 2022 (references to Fig 1 are not helping until the layout in Fig 1 is made more descriptive). . .

  See our comment on Figure 1 in the comments on section 2.

  but also, it turns out that these functional categories are not directly the categories used in Fig. 5, so why bother? Explain what is shown in Fig 5 without the confusion of the "functional categories". I also have no idea what the "NETCDF output" is part of: is it the I/O interface or is it the post-processing of the solution vector which ultimately would be a sub-category of the "CUAS-MPI solver "?

  There is, in fact an inconsistency between the labeling of Figure 1 w.r.t. the use of "The CUAS-MPI Solver". We will revise Figure 1 and be more precise which lines from the pseudocode in Figure 1 correspond to the functional categories we measured.

- l. 248: what does the "They" refer to here?

  We reformulated it.

- l. 267: consider rephrasing that first sentence (why using the past tense here? Maybe I am missing the point).

We will check.

- l. 272: the choice to output a solution here is odd, since this cost is obviously going to dominate the runtime for all coarser grids, based on the results from G600 of Fig 5.

See our previous comment on output at the start of this section.

- l. 276: I wouldn't say "profitably"

We think "efficiently" fits best in this context.

- Fig 7a: why no more log scale for the x axis of Fig 7a?

Good point — we will prepare Fig7a with a log scale for the revised version.

- Fig 7b and c are difficult to read and the accompanying text is not very easy to understand. Are we supposed to "read" them as going from right to left = increasing the number of MPI processes?

Indeed reading them from left to right would be a good way to draw conclusions from them, but similarly looking for minima of the individual curves is an approach. We understand that the current text is not sufficient for the reader and will improve the main text. This figure is actually most useful for planning simulations and as thus potential users do indeed need to be given a good explanation of this figure. Many thanks for pointing this out! Also why is the inflexion (breaking of scalability) starting for a higher number of cells per MPI processes for finer resolutions?

We will be more descriptive for this: finer grids have more cells in total than coarser grids. Thereby more processes are necessary to reach a lower number of cells per process. More processes cause more computational overhead, resulting in lower maximum throughput for finer grids.

- l. 279: "Smaller grids"= coarser resolution? (consider changing "larger grids" too)

We adopt the referee's suggestion.

- l. 288: what is the "sweet spot"?

In this context, we assume that the sweet spot is the minimum runtime and will clarify this in the text.

- l. 292: that last sentence needs rephrasing, not sure what is said here (what is "the spread"?)

We rephrased the sentence.

**2.2.7 Section 6**

I suggest merging part of this section with the Conclusions, and including the rest (the discussion about runtime and throughput) in the previous section.

We appreciate the suggestion, but in our view it is appropriate to differentiate between a more expensive discussion and precise formulation of the conclusion.

– l. 296: First sentence should be rewritten. Give specifics -what is high, highest?

We have included numbers for the resolution and have rephrased the sentence to 'The throughput shown by CUAS-MPI enables ice sheet wide simulations in high (600 m), but potentially not highest resolution (150 m) due to the computational costs.'

– What does "the code performs well" means ?

Yes, we agree 'the code performs well' is too vague as a statement. The intention here was to make the statement that one can compute longer time periods without having simulations that take months' to complete, seasonal simulations, which need hourly time steps and daily output, are taking too long in highest resolution. Highest resolution refers to the resolution of the bedrock topography, which is (currently!) 150 m. We are rephrasing the sentence in the revised version.

– What are the number of time steps required for a seasonal cycle?

We compute hourly time steps for the seasonal cycle but we write only daily output. This is of course a choice of the modeller, balancing available forcing data and affordability. It is worth to note, that seasonality does not end once the seasonal melt water availability vanishes, as the system still evolves beyond the end of the melt season.

– The requirements of 56 days for a 90 years run with 600m temporal res seems off, based on the results from Fig 7a. Please clarify. Also this is very much supercomputer-dependent and this should be specified.

This is indeed correct. We are into investigating this and will provide updated numbers in the sentence in the revised version. Many thanks for pointing this out!

– l. 301: what are "panGreenland simulations?

"panGreenland" → "Greenland-wide" - we will change this in the revised version.

– l. 302: what do you mean by that last sentence "The costs are also . . . "?

When we aim to run coupled simulations for projections, they are typically simulating 100 to 300 years. If we include a seasonal cycle of meltwater input, we end up with a daily coupling between ice sheet and hydrology models to capture the seasonal effect on the ice sheet, too. This is indeed costly, as Fig. 7 shows. This is what is meant here.

– What is the paragraph starting l. 307 doing here? It sounds like it would be better placed in the Introduction

Very well spotted - many thanks! We will move this to the introduction

– l. 313: "is" at the end of the sentence should be removed

We adopt the referee's suggestion.

– l. 315 "In out work, the next step . . . " → "The next step . . . "

We adopt the referee's suggestion.

- l. 316: the ISSM citation is odd, the website should be a footnote or a real citation altogether?

  We will apply according to the GMD guidelines

- l. 320: odd citation for preCICE too

  We will apply according to the GMD guidelines

- Paragraph starting l. 315 can be a perspective for, say, the Conclusions section but its placement here is odd. You could merge with the next paragraph but emphasis should be on the test cases results

  We agree on expanding the conclusion, but we prefer not to shorten the discussion.

- l. 326-333: what is said here ? Is it an extension of the paragraph starting l. 315?

  we will rephrase this

**2.2.8 Conclusions**

This section is too incomplete. It should either be augmented with the suggested paragraphs of Sect. 6, or merged with the discussion section altogether.

We believe it to be valuable to separate a discussion of the results from a concise description of our conclusions. However, our conclusion is, in fact, currently very short and we will expand it in the revised version.

- l. 349: CUAS-MPI is not a new code – or if it is it has not been demonstrated properly

  As we laid out above, CUAS-MPI is written from scratch and we will make this more clear in the revised version of the manuscript.

- l. 355-356: too vague - consider describing what these enhancements and model adaptation should entail. The last sentence should be impactful.

  Yes, we admit the last part of the conclusion is neither inspiring nor impactful. We will rephrase the conclusion for the revised version.

**2.2.9 References**

- The doi for the paper by Young, T. J., Christoffersen, P., Bougamont, M., and Stewart, C. L. should read https://doi.org/10.1073/pnas.2116036119

  We adopt the referee's suggestion.

**References**

450 Beyer, S., Kleiner, T., Aizinger, V., Rückamp, M., and Humbert, A.: A confined–unconfined aquifer model for subglacial hydrology and its application to the Northeast Greenland Ice Stream, The Cryosphere, 12, 3931–3947, https://doi.org/10.5194/tc-12-3931-2018, 2018.

Bindschadler, R. A., Vornberger, P. L., King, M. A., and Padman, L.: Tidally driven stick–slip motion in the mouth of Whillans Ice Stream, Antarctica, Annals of Glaciology, 36, 263–272, https://doi.org/10.3189/172756403781816284, 2003.

Engelhardt, H. and Kamb, B.: Basal sliding of Ice Stream B, West Antarctica, Journal of Glaciology, 44, 223–230,
455 https://doi.org/10.3189/S0022143000002562, 1997.

Ferris, J. G., Knowles, D. B., Brown, R. H., and Stallman, R. W.: Theory of Aquifer Tests, Tech. rep., U.S. Government Print. Office, https://doi.org/10.3133/wsp1536E, 1962.

Fischler, Y., Rückamp, M., Bischof, C., Aizinger, V., Morlighem, M., and Humbert, A.: A scalability study of the Ice-sheet and Sea-level System Model (ISSM, version 4.18), Geoscientific Model Development, 15, 3753–3771, https://doi.org/10.5194/gmd-15-3753-2022, 2022.

460 Hoffman, M. J., Catania, G. A., Neumann, T. A., Andrews, L. C., and Rumrill, J. A.: Links between acceleration, melting, and supraglacial lake drainage of the western Greenland Ice Sheet, Journal of Geophysical Research: Earth Surface, 116, https://doi.org/10.1029/2010JF001934, 2011.

Siegert, M. J.: Antarctic subglacial lakes, Earth-Science Reviews, 50, 29–50, https://doi.org/10.1016/S0012-8252(99)00068-9, 2000.

---

## Referee Report (RR1)

The authors have made considerable efforts to implement the requested changes from the first round of reviews. I believe that the clarity of the message is better conveyed by the restructuring of the manuscript and the overall English improvements. The introduction is now complete with a well written state of the art, and the place and originality of the present study -in particular, with respect to previous publications by Beyer et al. 2018, has been more explicitly emphasized throughout. The additional figures are helpful, and the occasional rework of existing ones appropriate. Section 3 is also much easier to follow, with a clearer description of the chosen test cases backed up with proper references.

I believe the quality of this preprint is very close to being suitable for publication; however, I would still recommend a couple of "major" improvements beforehand.

First, I find that Section 2.2 still suffers from vagueness; and contains superfluous information that would be better merged with the Appendix "workflow": from l. 154 to line 168, not a lot is said that is going to help in the understanding of the rest of the paper. Instead, and in preparation for the discussion of Section 4.3, I think this would be a good place to provide more details about the model's implementation. For instance, the sentence starting l.150 "We designed the grids and kernels of CUAS-MPI…" should be elaborated upon. Can you provide more details about these "grids" that you design ? Is it an original storage feature or does it rely entirely on PETSc? In the same fashion, the paragraph starting l. 177 could be more descriptive. You could also be more technical about what "CUAS-MPI system kernels" versus "CUAS-MPI system matrix" contain, and what PETSc feature they rely upon/use.

Indeed, being more descriptive in what you coded versus what PETSc is eventually in charge of would help make the analysis of Section 4.3 less generic: while I appreciate that CUAS-MPI is "an entirely different model simulating another compartment of the Earth system (than what is presented in Fischler et al., 2022)", I strongly disagree when the authors state that conclusions drawn from this previous paper do not apply here –unless they can provide a more detailed analysis of the culprit behind the observed throughput and scalability limits. An analysis culminating with the three paragraphs starting l. 347, incriminating the loss of performance "to the increasing communication overhead between MPI processes and decreasing computation performed on each MPI process" has to, at the very least, acknowledge similitudes with that previous work (Fischler et al., 2022) - which most likely also employed the same PETSc features and matrix/vector formatting. If anything else, to reinforce the statement made in Section 5 about how much improvements to the external libraries that you rely upon would improve your throughput in a multi-code environment (ie, ice sheet+hydrology) !

There are also a few small typos/"questions" which require the authors' attention before publication:
- Overall, the equations, citations and sections referencing are messy (lots of misplaced parenthesis, sometimes it is "Eq." sometimes "equation", sometimes "section" sometimes "Sec." …) . Hopefully this is something that the editor will correct prior to publication ? But it would be worth checking the Latex source file.

- The first sentence in the abstract is still a bit difficult to follow. I suggest rewriting it ("due to sliding …" → "through sliding and the location of lakes at the ice margin" ? "as well as the ocean circulation" → "it also impacts the ocean circulation …" ?)
- l.4 "on the ice sheet scale" → "**at** the ice sheet scale"
- l.7 "...the scaling behavior of…" → "the **strong** scaling"
- l.8 I would suggest completing the sentence with the type of supercomputers that you use
- In l. 31 the "inefficient system" is mentioned prior to introducing it. Perhaps you can add a descriptive term in parenthesis ?
- l. 39 missing "the" before head ?
- l. 44 "..., advanced for seasonal evolution of the hydrological system" is awkward, perhaps the sentence could be rephrased
- l. 46 "... both systems: Sommers et al evolves …" → "... both systems. Sommers et al. solve for the water …" (splitting the sentence)
- l. 49 "lead" → "led"
- l. 50/51 "as in Smith-Johnsen et al. …" → "such as in Smith-Johnsen et al. …, for example."
- I would suggest referring the reader to Section 2 for a definition of the greek symbols in the legend of Fig. 1
- l. 69 "... will benefit if simulations of the subglacial system are available for the same time period." → "... will benefit from simulations of the subglacial system for the same time period."
- l. 71 perhaps provide a number for "fine resolution" (100-500m ?)
- l. 72 "... software that uses parallel computers …" → "... softwares capable of using parallel computers …"
- The title 2.1 is still too short, please be more descriptive
- l. 107 why not use the acronym CUAS that you have introduced ?
- The numbers in Fig. 2, which will be very useful for Section 4, should be acknowledged in the caption of the figure
- l. 148 sub-glacial → subglacial
- the sentence starting l. 156 could be dropped altogether
- l. 177 "We" → "we" (no capital letter)
- l. 186 please provide a ref for MUMPS
- Fig 5: the exact meaning of "difference" should be given. Is it an absolute error ?
- The sentence starting l. 243 "To be able to compare performance …" is awkward and should be rewritten
- l. 245 "As linear solver, we use GMRES with a Jacobi preconditioner" → "We use the GMRES linear solver with a Jacobi preconditioner"
- l. 257 you talk about GXX runs prior to introducing the notation
- The first sentence of Section 4.1 is awkward, consider rephrasing
- l. 294 "sufficiently realistic" → "sufficiently realistically"
- l. 331 "The CUAS-MPI system kernels category, which, like the code categories discussed in the following, is also identified in Fig. 2, includes all kernels running on CUAS-MPI grids.The CUAS-MPI system kernels category contains the characteristics of the EPM, such as the confined-unconfined scheme, transmissivity change and flux, as two dimensional fields." → "**The four remaining functional parts are identified in Fig. 2.** In particular, the CUAS-MPI system kernels category

includes all kernels running on CUAS-MPI grids (see the blue box in Fig. 2)". This is just a rewrite suggestion
- l. 337 "the **preconditioned** GMRES implementation"
- l. 337 "Finally, NetCDF output includes …" → "Finally, the NetCDF output category includes …"
- l. 370 "fix" → "fixed"
- Figure 10a should feature the same x-axis legend as Figure 9: 96, 192, 384 etc. It is very confusing otherwise. Consider enlarging the police for both axis also, it looks like there is the space to do so.
- The paragraph of l. 375 is redundant and could be dropped
- l. 383: "98" → "96"
- Can you elaborate more on the sentence starting l. 389 ? What is the cause of needing more iterations and how does that number grow with resolution ? That is an interesting issue for further code improvement !
- When discussing potential runtime improvements (l. 446-459), it would be interesting to say a word about what you, within your code implementation, could be able to improve: perhaps communication overhead could be avoided by recomputing ghost cells less often ? Or could a different linear solver be potentially investigated - still within the PETSc framework ? It would bring much to the discussion.
- l. 479 I would be careful in saying this

---

## Author Response (AR2)

**GMD Reviews and Authors' Response concerning the paper "A parallel implementation of the confined-unconfined aquifer system model for subglacial hydrology: design, verification, and performance analysis (CUAS-MPI v0.1.0)"**

Yannic Fischler[1], Thomas Kleiner[2], Christian Bischof[1], Jeremie Schmiedel[4], Roiy Sayag[4], Raban Emunds[1,2], Lennart Frederik Oestreich[1,2], and Angelika Humbert[2,3]

[1]Department of Computer Science, Technical University Darmstadt, Darmstadt, Hesse, Germany
[2]Alfred-Wegener-Institut, Helmholtz-Zentrum für Polar- und Meeresforschung, Bremerhaven, Bremen, Germany
[3]Faculty of Geosciences, University of Bremen, Bremen, Germany
[4]Department of Environmental Physics, BIDR, Ben-Gurion University of the Negev, Sde Boker, Israel

**Correspondence:** Yannic Fischler (yannic.fischler@tu-darmstadt.de)

*Copyright statement.* ©2023 all rights reserved

**1 General Comments**

We thank the reviewer for his extensive work in the review process. We have tried to be more precise in our software section and adopted most of the minor issues raised.

**2 Review 1 by an anonymous referee**

**2.1 General**

The authors have made considerable efforts to implement the requested changes from the first round of reviews. I believe that the clarity of the message is better conveyed by the restructuring of the manuscript and the overall English improvements. The introduction is now complete with a well written state of the art, and the place and originality of the present study -in particular, with respect to previous publications by Beyer et al. 2018, has been more explicitly emphasized throughout. The additional figures are helpful, and the occasional rework of existing ones appropriate. Section 3 is also much easier to follow, with a clearer description of the chosen test cases backed up with proper references.

I believe the quality of this preprint is very close to being suitable for publication; however, I would still recommend a couple of "major" improvements beforehand.

15    First, I find that Section 2.2 still suffers from vagueness; and contains superfluous information that would be better merged with the Appendix "workflow": from l. 154 to line 168, not a lot is said that is going to help in the understanding of the rest of the paper. Instead, and in preparation for the discussion of Section 4.3, I think this would be a good place to provide more details about the model's implementation. For instance, the sentence starting l.150 "We designed the grids and kernels of CUAS-MPI..." should be elaborated upon. Can you provide more details about these "grids" that you design ? Is it an original

20    storage feature or does it rely entirely on PETSc? In the same fashion, the paragraph starting l. 177 could be more descriptive. You could also be more technical about what "CUAS-MPI system kernels" versus "CUAS-MPI system matrix" contain, and what PETSc feature they rely upon/use.

We now explicitly mention the PETSc features we use: Distributed Array, Matrix & Vector. However, we do not consider it appropriate to go into more detail. This paper is intended to present the parallel implementation of the confined-unconfined

25    aquifer system model. It is not intended to serve as PETSc tutorial. For readers familiar with PETSc and interested in these details, the source code, which is part of the submission package, provides the best explanation.

Indeed, being more descriptive in what you coded versus what PETSc is eventually in charge of would help make the analysis of Section 4.3 less generic: while I appreciate that CUAS-MPI is "an entirely different model simulating another compartment of the Earth system (than what is presented in Fischler et al., 2022)", I strongly disagree when the authors state that conclusions

30    drawn from this previous paper do not apply here –unless they can provide a more detailed analysis of the culprit behind the observed throughput and scalability limits. An analysis culminating with the three paragraphs starting l. 347, incriminating the loss of performance "to the increasing communication overhead between MPI processes and decreasing computation performed on each MPI process" has to, at the very least, acknowledge similitudes with that previous work (Fischler et al., 2022) - which most likely also employed the same PETSc features and matrix/vector formatting. If anything else, to reinforce the statement

35    made in Section 5 about how much improvements to the external libraries that you rely upon would improve your throughput in a multi-code environment (ie, ice sheet+hydrology)!

Indeed, the structure of performance studies is often similar. However, CUAS-MPI and ISSM are different codes with different properties, e.g.:

   – Discretization: 2D structured grid versus 3D unstructured grid

40    – Physical model: Groundwater flow equation versus Stokes equation

   – Numerics: Finite differences versus finite element

CUAS-MPI and ISSM both "use PETSc", but they use different PETSc data structures of this extensive software toolkit. As a result, they also have different bottlenecks: In ISSM we have seen the matrix assembly as the most significant scaling bottleneck, while in CUAS-MPI it is the grid boundary exchange. A second major difference is the fact that the linear solver is

45    negligible in the performance of ISSM, but in CUAS-MPI it is of significant cost.

In this paper, we present a new simulation tool and the scaling study is intended to help scientists to determine which experiments they can afford with the CUAS-MPI implementation. While we do not compare scaling issues of CUAS-MPI

versus ISSM, we do want to set the throughput of CUAS-MPI in relation to that of ISSM as both simulations might be used to run joint ice shield and hydrology setups.

The comparison of the scaling of different PETSc implementations may be a useful study, but it is not the focus of this work. In particular, if one were to undertake such a study, the factors deciding the choice of software investigated would likely be based on their usage of PETSc features, not their application domain.

**2.2 Other**

There are also a few small typos/"questions" which require the authors' attention before publication:

- Overall, the equations, citations and sections referencing are messy (lots of misplaced parenthesis, sometimes it is "Eq." sometimes "equation", sometimes "section" sometimes "Sec." ...). Hopefully this is something that the editor will correct prior to publication? But it would be worth checking the Latex source file.
  Thank you. We have put effort into finding all odd usages.

- The first sentence in the abstract is still a bit difficult to follow. I suggest rewriting it ("due to sliding ..." → "through sliding and the location of lakes at the ice margin"? "as well as the ocean circulation" → "it also impacts the ocean circulation ..."?)
  Many thanks. There are four individual effects listed: (1) sliding, (2) location of lakes at the ice sheet margin, (3) freshwater input into the ocean from beneath the ice sheet at the grounding line and (4) freshwater input into the ocean arising from subglacial outflow over land. We have changed 'due to' in 'through' and have added (i)-(iv) to emphasise the different contributions better.

- l.4 "on the ice sheet scale" → "at the ice sheet scale"
  Done.

- l.7 "...the scaling behavior of..." → "the strong scaling"
  We prefer to keep our formulation as we think introducing the term "the strong scaling" without further context is confusing, and just "scaling" is general enough for an abstract.

- l.8 I would suggest completing the sentence with the type of supercomputers that you use
  We now mention the supercomputer we use in the abstract.

- In l. 31 the "inefficient system" is mentioned prior to introducing it. Perhaps you can add a descriptive term in parenthesis? We use the term "inefficient" already in l. 29 (first revised version) and now provide a reference to Fig. 1 additionally.

- l. 39 missing "the" before head?
  We adopt the referee's suggestion.

– l. 44 ”..., advanced for seasonal evolution of the hydrological system” is awkward, perhaps the sentence could be rephrased

Indeed! We have rephrased to: "De Fleurian et al. (2014) use a double continuum approach with two different porous layers, one for the distributed system and one for the efficient system, with the second being important for the seasonal evolution of the hydrological system (de Fleurian et al., 2016)."

– l. 46 "... both systems: Sommers et al evolves . . . " → "... both systems. Sommers et al. solve for the water . . . " (splitting the sentence)

Done.

– l. 49 "lead" → "led"

Done.

– l. 50/51 "as in Smith-Johnsen et al. . . . " → "such as in Smith-Johnsen et al. . . . , for example."

Done.

– I would suggest referring the reader to Section 2 for a definition of the greek symbols in the legend of Fig. 1

Thank you. A reference is added as suggested.

– l. 69 "... will benefit if simulations of the subglacial system are available for the same time period." → "... will benefit from simulations of the subglacial system for the same time period."

Done.

– l. 71 perhaps provide a number for "fine resolution" (100-500m?)

– l. 72 "... software that uses parallel computers . . . " → "... softwares capable of using parallel computers . . . "

We adopt the referee's suggestion.

– The title 2.1 is still too short, please be more descriptive

We use "Description of the physical Model"

– l. 107 why not use the acronym CUAS that you have introduced?

We have dropped "used in the confined-unconfined aquifer" altogether.

– The numbers in Fig. 2, which will be very useful for Section 4, should be acknowledged in the caption of the figure

Thank you. We have changed the figure caption accordingly. "The numbers 1–5 in white circles are used again in Sec. 4, e.g. Fig. 8, for cross-referencing. "

– l. 148 sub-glacial → subglacial

Done.

– the sentence starting l. 156 could be dropped altogether

We see it worse to mention and moved it to appendix.

– l. 177 "We" → "we" (no capital letter)

Done.

– l. 186 please provide a ref for MUMPS

We adopt the referee's suggestion.

– Fig 5: the exact meaning of "difference" should be given. Is it an absolute error?

Thank you. We agree with the reviewer and change the caption of Fig. 5 to exactly describe the meaning of "difference".

Figure 5. ...(Bottom) The difference between the computed and predicted drawdown, shown in the top panel, indicates an overall small discrepancy between the numerical and theoretical values.

– The sentence starting l. 243 "To be able to compare performance ..." is awkward and should be rewritten

We now use: "We run the model for 24 time steps (one model day) using different grid resolutions to compare the model performance and scaling behavior."

– l. 245 "As linear solver, we use GMRES with a Jacobi preconditioner" → "We use the GMRES linear solver with a Jacobi preconditioner"

We adopt the referee's suggestion.

– l. 257 you talk about GXX runs prior to introducing the notation Thank you. We now state the resolution (1200 m and 2400 m) explicitly here and keep the introduction of the "GXX" in section 4.1.

– The first sentence of Section 4.1 is awkward, consider rephrasing

We changed this to: "The model domain consists of a rectangular area containing the Greenland ice sheet. Grid points of the hydrological system for which Eq. 1 is solved are colored red in Figure 6."

– l. 294 "sufficiently realistic" → "sufficiently realistically"

We adopt the referee's suggestion.

– l. 331 "The CUAS-MPI system kernels category, which, like the code categories discussed in the following, is also identified in Fig. 2, includes all kernels running on CUAS-MPI grids.The CUAS-MPI system kernels category contains the characteristics of the EPM, such as the confined-unconfined scheme, transmissivity change and flux, as two dimensional fields." → "The four remaining functional parts are identified in Fig. 2. In particular, the CUAS-MPI system kernels category includes all kernels running on CUAS-MPI grids (see the blue box in Fig. 2)". This is just a rewrite suggestion

Thank you. We made the change accordingly.

- l. 337 "the preconditioned GMRES implementation"

  We adopt the referee's suggestion.

- l. 337 "Finally, NetCDF output includes ..." → "Finally, the NetCDF output category includes ..."

  We adopt the referee's suggestion.

- l. 370 "fix" → "fixed"

  Done.

- Figure 10a should feature the same x-axis legend as Figure 9: 96, 192, 384 etc. It is very confusing otherwise. Consider enlarging the police for both axis also, it looks like there is the space to do so. Thank you. That is a good suggestion. We consider applying this change in the final version of the figures.

- The paragraph of l. 375 is redundant and could be dropped

  We consider it relevant to make this point, even if it could be deduced from previous data.

- l. 383: "98" → "96"

  Done.

- Can you elaborate more on the sentence starting l. 389? What is the cause of needing more iterations and how does that number grow with resolution? That is an interesting issue for further code improvement

  This is of course of practical importance for simulation planning, and we elaborate on the implications in the text (l. 390 – 401, first revised version). We don't think this would be worth another paragraph or figure. Especially as we expect this to be related to the model set-up and not the model implementation. On a higher resolution grid, more and more features of the model domain can be resolved, and thus, the gradients in the geometry might increase (bed elevation, ice thickness) as well as the complexity of the lateral boundary conditions, e.g. some outflow boundary conditions (small fjords) are only resolved on a higher resolution grid. We have not investigated those effects and prefer just to state our findings from the performance measurements we did. The set-ups we have chosen for the scaling analysis are on purpose based on the geometries that would use in science related to hydrology. The drawback is that we do not solve the same physical system using different resolutions as explained above.

- When discussing potential runtime improvements (l. 446-459), it would be interesting to say a word about what you, within your code implementation, could be able to improve: perhaps communication overhead could be avoided by recomputing ghost cells less often? Or could a different linear solver be potentially investigated - still within the PETSc framework? It would bring much to the discussion.

  We already state out, that we think, that hybrid parallelism is a major potential, because the grid computation will be more efficient on shared memory, than the current distributed memory + communication. To use this a hybrid parallel solver is necessary as well. We have not elaborated on this yet, but it has potential.

  We additionally state out, improvement potential of PETSc: using persistent MPI communication.

– l. 479 I would be careful in saying this

170 We agree. Given the difficulties in observing the subglacial hydrological system, this sentence probably sounds overconfident. We now write: "In contrast to the Greenland Ice Sheet, the hydrology below the Antarctic ice sheet is expected to be driven by basal water production without additional water from the ice sheet surface draining to the base during the summer season."